



# Statistical characteristics of raindrop size distribution over Western Ghats of India: wet versus dry spells of Indian Summer Monsoon

*Uriya Veerendra Murali Krishna[1], Subrata Kumar Das[1*], Ezhilarasi Govindaraj Sulochana[2], Bhowmik Utsav[1], Sachin Madhukar Deshpande[1], and Govindan Pandithurai[1]*

[1]Indian Institute of Tropical Meteorology, Ministry of Earth Sciences, Pashan, Pune-411008, India

[2]College of Engineering, Guindy, Chennai-600025, India

[*]Correspondence to Subrata Kumar Das (skd_ncu@yahoo.com).



**Abstract:**

The nature of raindrop size distribution (DSD) is analyzed during wet and dry spells of the

Indian Summer Monsoon (ISM) in the Western Ghats (WGs) region by using Joss-Waldvogel

Disdrometer (JWD) measurements. The observed DSDs are fitted with gamma distribution, and the

DSD characteristics are studied during ISM season (June-September) of 2012-2015. The DSD spectra

show distinct diurnal variation during the wet and dry spells. The dry spells exhibit a strong diurnal

cycle with two peaks, while the diurnal cycle is not so prominent in the wet spells. Results reveal the

microphysical characteristics of warm rain during both the wet and dry periods. Even though the warm

rain processes are dominant in the WGs region, the underlying dynamical processes cause the

differences in DSD characteristics during the wet and dry spells. In addition, the differences in DSD

spectra with different rain rates are also observed. The DSD spectra are further analyzed by separating

into stratiform and convective types. Finally, an empirical relationship between the slope parameter, $\Lambda$

and shape parameter, $\mu$ is derived by best fitting the quadratic polynomial for the observed data during

both wet and dry spells as well as for the stratiform and convective types of rain. The $\mu$-$\Lambda$ relations

obtained in the present study are slightly different in comparison with the previous studies.

**Keywords:** Raindrop size distribution, Wet and dry spells, Monsoon, Western Ghats, Disdrometer.



# 1. Introduction

Western Ghats (WGs) is one of the heavy rainfall regions in India. WGs receives a large amount of rainfall (~6000 mm) during the Indian Summer Monsoon (ISM) period (Das et al., 2017, and references therein). Shallow clouds contribute significantly to the monsoon rainfall on the windward side (Kumar et al., 2013; Das et al., 2017; Utsav et al., 2017, 2019) and deep convection in the leeward side (Utsav et al., 2017, 2019; Maheskumar et al., 2014) of the WGs. In addition, thunderstorms also occur over WGs. However, they are very few during the monsoon period. The rainfall distribution in the WGs region is complex in which topography plays a significant role (Houze et al., 2012, and references therein). The distribution of rainfall on the WGs region depends on the area, whether it is on the windward side or leeward side of the mountains. These different properties correspond to different physical mechanisms. The intense rainfall in the windward side of the mountains, usually called the orographic precipitation comes from shallower clouds with long-lasting convection (Das et al., 2017; Utsav et al., 2019). One of the significant issues in precipitation measurements in the WGs region is the unavailability of a stable platform.

The ISM shows large spatial and temporal variability. It is known that during the active (with a high amount of rainfall) and break (with a little or no rain) spells of the ISM, there are different behaviours in the formation of weather systems and large-scale instability. The strength of the ISM rainfall depends on the frequency and duration of active and break spells (Kulkarni et al., 2011). This intra-seasonal oscillation of precipitation is considered as one of the most critical sources of weather variability in the Indian region (Hoyos and Webster, 2007). From the earlier studies of Ramamurthy (1969), active and break spells of the ISM have been extensively studied, especially during the last two





decades (e.g., Goswami and Ajaya Mohan, 2001; Gadgil and Joseph, 2003; Uma et al., 2011; Rajeevan et al., 2012; Mohan and Rao, 2012; Das et al., 2013; Rao et al., 2016). The characteristic features of ISM active and break spells have been well understood; for example, their identification (Rajeeven et al., 2006; Rajeevan et al., 2010), spatial distribution (Ramamurthy, 1969; Rajeevan et al., 2010),

circulation patterns (Goswami and Ajaya Mohan 2001; Rajeevan et al., 2010), vertical wind and thermal structure (Uma et al., 2011), rainfall variability (Deshpande and Goswami, 2014; Rao et al., 2016) and the macro- and micro-physical features of clouds (Rajeevan et al., 2012; Das et al., 2013). Even though different dynamical mechanisms for the observed rainfall distribution during the wet and dry spells of ISM are well understood, the investigation on microphysical processes for rain formation is still

lacking.

Raindrop size distribution (DSD) is a fundamental microphysical property of the precipitation. The DSD characteristics are related to processes such as hydrometeor condensation, coalescence, and evaporation. These are important parameters affecting the microphysical processes in the parameterization schemes of the numerical weather prediction models (Gao et al., 2011). Hence,

numerous observations of DSD during different types of precipitation, different seasons, and different intra-seasonal periods at different locations are essential for better representation of physical processes in the parameterization schemes. As a result, the numerical weather prediction model communities are continuing their efforts to improve the simulation of clouds and precipitation at the monsoon intra-seasonal scales by better representing the microphysical processes through parameterization schemes.

Different DSD characteristics lead to different reflectivity ($Z$) and rainfall rate ($R$) relations. Hence, understanding the variability in DSD is vital to improve the reliability and accuracy in the quantitative


precipitation estimation from radars and satellites (Rajopadhyaya et al., 1998; Atlas et al., 1999; Viltard et al., 2000; Ryzhkov et al., 2005).

The active and break spells in the WGs region are nearly identical with the active and break

phases over the core monsoon zone (Gadgil and Joseph, 2003). The distribution of convective clouds in the WGs region exhibit distinct spatiotemporal variability at intra-seasonal time scales (wet: analogous to an active period of ISM and dry: similar to a break period of ISM) during the ISM. Utsav et al. (2019) studied the characteristics of convective clouds over WGs using X-band radar observations along with European Center for Medium-range Weather Forecasting (ECMWF) interim reanalysis

(ERA-Interim), and Tropical Rainfall Measuring Mission (TRMM) satellite datasets. Their study revealed that the wet spells are associated with negative geopotential height anomalies at 500 hPa, negative outgoing long-wave radiation (OLR) anomalies, and positive precipitable water anomalies. All these features promote the anomalous south-westerlies, which favours the growth of convective elements over WGs. In contrast, positive geopotential height anomalies, positive OLR anomalies, and

negative precipitable water anomalies are observed during the dry spells. These atmospheric conditions suppress the convective activity in the Arabian Sea, and hence little to no rain is seen over WGs during the dry periods. These different dynamical properties affect the convection during the wet and dry spells over WGs. However, the DSD (often used to infer the microphysical processes of rain) during the wet and dry periods of ISM are least addressed, especially in the WGs region.

Climatological studies of DSD at several locations in a given region are rare, especially in the WGs region. A few attempts have been made to understand the DSD characteristics in the WGs. For example, Konwar et al. (2014) studied the DSD characteristics by fitting three-parameter gamma

function during the monsoon season. They observed that bimodal and monomodal DSD during low and high rainfall rates, respectively. However, their study is limited to brightband and non-brightband conditions only. Harikumar (2016) studied the differences between DSD on the coastal (Kochi) and high altitude (Munnar) station located in the WGs region. He found for a given rain rate, more number of larger size drops are present at Munnar than at Kochi. Das et al. (2017) studied the DSD characteristics during different precipitating systems in the WGs region using Disdrometer and Micro Rain Radar measurements. They noticed different *Z-R* relations during different types of precipitation. Sumesh et al. (2019) studied the DSD differences between mid- (Braemore, 400 m above MSL) and high-altitude (Rajamallay, 1820 m above MSL) regions in southern WGs during brightband events. They observed bimodal DSD in mid-altitude station and monomodal DSD in the high altitude station. Their study also confined to stratiform rain only.

There are limited studies of DSDs exist in the WGs region by considering long-term dataset. This work is the first study to analyze the DSD characteristics by considering the monsoon intra-seasonal oscillations (wet and dry spells). The present study brings out the results of a unique opportunity by analyzing a more extensive dataset and also considering the different phases of the monsoon intra-seasonal oscillations in the WGs. With this background, the current study attempted to address the following issues:

1. How do the DSD characteristics vary during wet and dry spells in the WGs region?

2. Does the wet and dry spell rainfall have different microphysical origin over the complex terrain of WGs?





3. Does the DSD show any diurnal differences like rainfall distribution during wet and dry spells over WGs?

4. Establish the best fit for $\mu$-$\Lambda$ relationships during wet and dry spells.

The paper is organized as follows: the details of the instrument and dataset used are presented in section 2. The methodology adopted for the separation of rainy days into wet and dry spells is given in section 3. A brief overview of the DSD variation with topography is in section 4. The observational results of DSDs during the wet and dry spells and the possible reasons are reported in section 5. The summary of this study is provided in section 6.

## 2. Instrument and Datasets

Four years (2012-2015) Joss-Waldvogel Disdrometer (JWD) measurements during the monsoon months (June to September) at the High Altitude Cloud Physics Laboratory (HACPL; located in the windward slopes of the WGs), Mahabaleshwar (17.92°N, 73.6°E, ~1.4 km above mean sea level) in the WGs is utilized to understand the DSD variations during the wet and dry spells of ISM.

The JWD is an impact type disdrometer, which measures the hydrometeors with sizes ranging from 0.3 to 5.1 mm and arranges them in 20 channels (Joss and Waldvogel, 1969). The JWD has a sensor to estimate the diameters of hydrometeors. Once the hydrometeors hit the 50 cm$^2$ styrofoam cone, the voltage is induced by the downward displacement, which is directly correlated with the drop size. The accuracy of JWD is 5% of the measured drop diameter. Although the JWD is generally accepted to be the standard instrument for DSD measurements (Tokay et al., 2005), it has several shortcomings, such as noise, sampling errors, and wind, etc. (Tokay et al., 2001; Tokay et al., 2003). In



addition to the above shortcoming, the JWD miscounts raindrops in the lower size bins, specifically for

drop diameters below 1 mm (Tokay et al., 2003). An effort has been made to overcome this deficiency

by discarding noisy measurements and applying the error correction matrix provided by the

manufacturer. To reduce the sampling error arising due to insufficient drop counts at lower rain rates,

the rain rates less than 0.1 mm hr$^{-1}$ are discarded in the present study. During heavy rain, the JWD

underestimates the number of smaller drops, known as disdrometer dead time. To account the

aforementioned error in the JWD estimates, the rain rates during wet and dry spells are analyzed. It is

observed that ~85% (90%) of the rain rates lies below 8 mm hr$^{-1}$ during wet (dry) spells (figure not

shown). By using the noise-limit diagram of Joss and Gori (1976), Tokay et al. (2001) investigated the

underestimation of small drops by JWD. They found that 50% of the drops below 0.4 mm cannot be

detected by the JWD when the rainfall rate is above 20 mm hr$^{-1}$. In the present study, only 4% (1%) of

the rain rates exceed 20 mm hr$^{-1}$ during wet (dry) spells. Hence, the underestimation of small drops by

JWD is negligible the study region. Tokay et al. (2001) further demonstrated that the gamma parameters

(such as normalized intercept parameter, rain rate, etc.) derived from long-term observations by JWD

and two-dimensional video disdrometer (2DVD) are in good agreement. In the present study, we

examined the DSD differences between wet and dry spells of the ISM using long-term (four seasons for

4 years) dataset. So it is appropriate to consider the undercounting of small drops may not affect much

the gamma DSD. Further, the underestimation of smaller drops for higher rain rate (4% for wet spells

and 1% for dry spells) may not affect the conclusion as this work does not intend to quantify the DSD

variations. Instead, it aims to understand the DSD variability during wet and dry spells over the

complex terrain. Further, there is no consensus regarding the JWD sampling period. The undersized





integration period can contribute to numerical fluctuations in DSDs, whereas higher sampling time may

miscount actual physical deviations (Testud et al., 2001). Hence, in the present study, we have averaged

the JWD measurements into 1 min period to filter out these deviations.

The concentration of raindrops, $N(D)$ (mm$^{-1}$ m$^{-3}$) at an instant of time is

$$N(D) = \sum_{i=1}^{20} \frac{n_i}{A \, \Delta t \, v(D_i) \, \Delta D_i} \tag{1}$$

where $A$ is the surface area of observation (50 cm$^2$), $t$ is the integration time, $n_i$ is the number of

raindrops in the size class $i$, and $D_i$ is the mean diameter of size class $i$. $v(D_i)$ is the terminal velocity of

the raindrop in $i$ channel and is estimated from Gunn and Kinzer (1949) as

$$v(D_i) = 9.65 - 10.3 \, e^{-6 \, D_i} \tag{2}$$

The rain rate $(R)$ and reflectivity $(Z)$ are estimated by assuming that the momentum is entirely

due to the terminal fall velocity of the raindrops and the raindrops are spherical and assume Rayleigh

scattering and expressed as

$$R = \frac{\pi}{6} \frac{3.6}{10^3} \frac{1}{A \times t} \sum_{i=1}^{20} (n_i \, D_i^3) \tag{3}$$

$$Z = \sum_{i=1}^{20} N(D_i) D_i^6 \, \Delta D_i \tag{4}$$

The one-minute DSD measurements obtained from JWD are fitted with a three-parameter

gamma distribution, as suggested by Ulbrich (1983). The details about the DSDs used in the present

study can be found in Das et al. (2017) and Krishna et al. (2017).

The functional form of the gamma distribution assumed for the DSD is expressed as





$$N(D) = N_0 D^\mu exp\left[-(3.67 + \mu)\frac{D}{D_0}\right] \qquad (5)$$


where, $N(D)$ is the number of drops per unit volume per unit size interval, $N_0$ (in $m^{-3}$ $mm^{-(1+\mu)}$) is the number concentration parameter, $D$ (in mm) is the drop diameter, $D_0$ (in mm) is the median volume diameter, and $\mu$ (unitless) is the shape parameter (Ulbrich, 1983; Ulbrich and Atlas, 1984). The gamma DSD parameters are calculated using moments proposed by Cao and Zhang (2009). Here, $2^{nd}$, $3^{rd}$, and

$4^{th}$ moments are utilized to estimate the Gamma parameters. This method gives relatively fewer errors compared to other methods (Konwar et al., 2014). The '$n$' order moment of the distribution can be calculated as

$$M_n = \int_0^\infty D^n \ N(D) \ dD \qquad (6)$$

The shape parameter, $\mu$, and the slope parameter, $\Lambda$ are given by


$$\mu = \frac{1}{(1-G)} - 4 \qquad (7)$$

$$\Lambda = \frac{M_2}{M_3}(\mu + 3) \qquad (8)$$

Where

$$G = \frac{M_3^2}{M_2 \ M_4} = \frac{\left[\int_0^\infty D^3 \ N(D) \ dD\right]^2}{\left[\int_0^\infty D^2 \ N(D) \ dD\right]\left[\int_0^\infty D^4 \ N(D) \ dD\right]} \qquad (9)$$



The other parameters, normalized intercept parameter, $N_w$ (in mm$^{-1}$ m$^{-3}$), mass-weighted mean

diameter, $D_m$ (in mm), and liquid water content, $LWC$ (in gm m$^{-3}$), are calculated following Bringi and

Chandrasekar (2001).

$$D_m = \frac{\int_0^\infty D^4 \, N(D) \, dD}{\int_0^\infty D^3 \, N(D) \, dD} \qquad (10)$$

$$LWC = 10^{-3} \frac{\pi}{6} \rho_w \int_0^\infty D^3 \, N(D) \, dD \qquad (11)$$

$$N_w = \frac{4^4}{\pi \, \rho_w} \left( \frac{10^3 \, LWC}{D_m^4} \right) \qquad (12)$$

Apart from JWD, the ERA-Interim (Dee et al., 2011) dataset is also used to understand the

dynamical properties responsible for different DSD characteristics during wet and dry spells. The ERA-

Interim provides atmospheric data on 60 levels in the vertical from the surface to 0.1 hPa. The ERA-

Interim data are available at 3-hourly and 6-hourly intervals. In the present study, temperature (K), and

specific humidity (kg kg$^{-1}$) at 700 hPa with a spatial resolution of 0.25º × 0.25º at 0000 UTC are

considered during ISM of 2012-2015. The specific humidity at 700 hPa infers the amount of water

vapour available for the cloud formation over the study region, WGs.

The daily accumulated rainfall collected by the India Meteorological Department (IMD) rain

gauge is used to identify the wet and dry spells of ISM. The IMD receives the rainfall accumulations at

08:30 LT (LT=UTC+05:30 hrs) every day. To examine the JWD data quality, the daily accumulated

rainfall measured by the JWD is compared with the daily accumulated rainfall collected from the rain



gauge. For comparison, JWD rainfall data accumulated at 08:30 LT is calculated for all the days during

the monsoon season of 2015. The daily accumulated rainfall collected by rain gauge and JWD above 1

mm is considered for the comparison. A total of 76 days of data is utilized. The non-availability of data

for this period may occur either due to maintenance activity or due to non-rainy days. Figure 1 shows

the scattered plot of daily accumulated rainfall between JWD and rain gauge. A linear fit is carried out

to the scatter plot and is displayed with the grey line in the figure. The correlation coefficient is about

0.99 between the two measurements despite their diverge physical and sampling characteristics. The

bias in JWD measured rainfall is about -0.7 mm, and root mean square error is about 2.9 mm. These

results suggest that the JWD measurements can be utilized to understand the DSD characteristics during

the wet and dry spells in the WGs region.

**3. Identification of wet and dry spells**

In the present study, an objective methodology proposed by Pai et al. (2014) is used to identify

the wet and dry spells. The IMD generated high-resolution gridded rainfall data using a rain gauge

network over the Indian region. High-resolution ($0.25^o \times 0.25^o$) daily gridded IMD rainfall dataset is

utilized for 32 years (1979-2011) over Mahabaleshwar ($17.75^o$N-$18^o$N and $73.5^o$E-$73.75^o$E), grid to

identify the wet and dry spells. The area-averaged daily rainfall time series is constructed for this region

during the monsoon period (1$^{st}$ June to 30$^{th}$ September) for the four years (2012-2015) as well as the

monsoon period for the 32 years data. For a given monsoon period, the difference of daily average

rainfall for four seasons and the daily average of long-term data provides the daily anomalies. The

standard deviation of daily average rainfall is calculated from 32 years of rain gauge data from IMD.



The standardized anomaly time series is obtained by normalizing the daily anomalies with the corresponding standard deviations.

$$Events = \frac{(Av.\,of\,daily\,rain - Av.\,of\,long\,term\,rain)}{St.dev.\,of\,daily\,rain} \qquad (13)$$

These standardized anomaly time series are used to separate the wet and dry spells. A period in this standardized anomaly time series is marked as wet (dry) if the standardized anomaly exceeded a value of 0.5 (-0.5) for consecutive three days or more (Utsav et al., 2019). Figure 2 shows the standardized rainfall anomalies calculated using eq. (13). Table 1 shows the number of wet and dry days during the study period. It is observed that there is more number of dry days during 2012-2015 monsoon seasons, and July has comparatively more number of wet days. In this work, 44,640 (149,760) 1-min raindrop spectra are analyzed during the wet (dry) days for 2012-2015 of ISM.

## 4. DSD overview-Topographic perspective:

The single point-wise instrument is not sufficient to address the orographic impacts on DSD characteristics. One of the difficulties in studying the effect of orography on DSD properties is the unavailability of many disdrometers deployed in the windward and leeward sides of the WG, which could capture the topography variations across the WGs region. However, in the present study, an overview of the DSD characteristics are presented on the windward and leeward sides of the WGs by using the Global Precipitation Measurement (GPM) mission satellite products. The GPM level 3 data provides different DSD parameters like $D_m$ and $N_w$ at a spatial resolution of $0.25^{\text{o}} \times 0.25^{\text{o}}$ from $60^{\text{o}}$S to $60^{\text{o}}$N. The GPM is the first space-borne dual-frequency precipitation radar (DPR) contains Ku band at



13.6 GHz and Ka-band at ~35.5 GHz. The details of the satellite mission can be found in Huffman et al. (2015), and the dataset used in the present analysis can be found in Krishna et al. (2017).

The GPM-DPR estimate $D_m$, and $N_w$ using the dual-frequency ratio (DFR) method. However,

the GPM-DPR suffers limitations. The DSD parameterization used in the GPM-DPR is the gamma distribution with a constant shape parameter, $\mu$=3 (Liao et al., 2014). The constant value of '$\mu$' introduces errors in the retrievals. The retrieval of $D_m$ using the DFR method is iterative, and the $D_m$ has two solutions when the DFR is less than 0 (Meneghini et al., 1997; Liao et al., 2003; Mardiana et al., 2004). The uncertainties in the GPM-DPR in estimating the DSD are detailed in Seto et al. (2013), Liao

et al. (2014), etc. Recently, Krishna et al. (2017) assessed the DSD measurements from GPM in the WGs region by comparing them with the ground-based disdrometer. They showed that the seasonal variations in $D_m$ and $N_w$ are well represented in the GPM measurements. However, they underestimate $D_m$ and $N_w$ value in comparison to the ground-based disdrometer measurement. Radhakrishna et al. (2016) also found that the GPM underestimates (overestimates) the mean $D_m$ ($N_w$) during the southwest

and northeast monsoons over Gadanki, a semiarid region of India. They showed that the single-frequency algorithm underestimates the mean $D_m$ by ~0.1 mm below 8 mm hr$^{-1}$, and the underestimation is a little higher at higher rain rates. Whereas in the dual-frequency algorithm, the mean $D_m$ is nearly the same below 8 mm hr$^{-1}$ but underestimates (~0.1 mm) at higher rain rates. Further, the underestimation is very small for $D_m$ values below 1.5 mm. In the present study, most of the $D_m$ values

present below 1.5 mm. Hence, it is reasonable to consider the GPM measurements to have an overview of DSD characteristics over the WGs.





Three locations are selected to understand the rain microphysical processes at different topographic regions in WGs. These locations are the ocean, high altitude cloud physics laboratory (HACPL; located on the windward slope of the WGs), and leeward side of the WGs. The DSD

differences in these three sites can partially infer the effect of orography on DSD. Figure 3 shows the distribution of $D_m$ over the ocean, windward, and leeward sides of the WGs. In this plot, the box represents the data between first and third quartiles, and the whiskers show the data from 12.5 and 87.5 percentiles. The horizontal line within the box represents the median value of the distribution. The distribution of $D_m$ is smaller over the ocean and high altitude site, whereas the $D_m$ shows large

variability on the leeward side. Further, the median value of $D_m$ is low over the ocean compared to the windward and leeward sides of the mountain. The smaller distribution of $D_m$ over the ocean and high altitude site can be attributed to the predominance of shallow clouds/cumulus congestus. In addition, the lower median $D_m$ represents the shallow convection over the ocean. The broader distribution and relatively higher median value of the $D_m$ represent the continental convection over the leeward side of

the mountains. Zagrodnik et al. (2019) also observed narrow $D_m$ distribution during the Olympic Mountains Experiment (OLYMPEX) on the windward side of the Olympic peninsula. Similarly, the large variability in $D_m$ on the leeward side of the mountain represents the presence of deeper clouds.

## 5. Results and Discussion

The DSD and rain integral parameters during the wet and dry spells are examined in terms of diurnal and with different types of precipitation (convective and stratiform). In this study, the raindrops



with diameters less than 1 mm are considered as small drops, with diameters in the range 1-4 mm are regarded as mid-size drops and with diameters above 4 mm are considered as large drops.

### 5.1. Raindrop size distribution during wet and dry spells

The information on the background microphysical processes, which are responsible for precipitation formation in convective and stratiform systems, could be inferred from observed variations in the DSDs at the ground. Figure 4 shows the temporal evolution of normalized raindrop concentration during wet and dry spells, exhibiting distinct diurnal features. The concentration of smaller drops (Figure 4a) is higher during the dry periods. The higher concentration of small drops in dry spells

indicates the predominance of orographic convection over WGs. In the mountain regions, DSDs evolved through warm/shallow rain processes. This warm rain is produced when the upslope wind is stronger, and moisture availability is high (White et al., 2003). In such a situation, the strong orographic wind enhances the growth of cloud droplets via condensation, collision, and coalescence (Konwar et al., 2014). Further, a large number of small raindrops during the dry spells indicate that the breakup and

evaporation processes may be more efficient during the dry periods. In the smaller drop spectra, dry spells exhibit a strong diurnal cycle with a primary maximum in the afternoon hours (1500-1900 LT) and a secondary peak in the night time (2300-0500 LT). This diurnal feature is also noted by Utsav et al. (2019) in the 15-dBZ echo top height (ETH) from X-band radar observations during the dry spells. However, such a diurnal cycle is not present in smaller drops during the wet spells. These smaller drops

show a little higher concentration during morning hours (0500-0700 LT), representing the oceanic nature of rainfall (Rao et al., 2009; Krishna et al., 2016).





In the mid-size drops (Figure 4b), the concentration is higher in wet spells compared to dry spells. The higher concentration of mid-size drops during the wet spells are due to the collision-coalescence process (Rosenfeld and Ulbrich, 2003), and accretion of cloud water by raindrops (Zhang et
al., 2008). This result indicates that the congestus clouds are omnipresent during the wet spells. Further, in the mid-size drops, both the spells exhibit a diurnal cycle; however, their strengths are different. The wet spells exhibit two broad maxima, one in the late afternoon (1400-1900 LT) and the other in the early morning (0500-0700 LT) times. The dry spells also show two maxima, one in the late afternoon (1400-1900 LT) as in the wet periods, and the other in the night time (2300-0500 LT). Such a diurnal
cycle is also observed in rainfall features over WGs (Shige et al., 2017; Romatschke and Houze, 2011). Shige et al. (2017) found a continuous rainfall with a double-peak structure of nocturnal and afternoon-evening maxima in the WGs region. Romatschke and Houze (2011) observed a double peak rainfall pattern in the WGs region. They proposed that the morning peak is related to oceanic convection while the afternoon peak is associated with the continental convection.

Figure 5 shows the mean DSDs during wet and dry spells along with the seasonal mean DSD for the study period. Here, *N(D)* is plotted on a logarithmic scale to accommodate its large variability. In general, the DSDs during the dry spells are narrower than the DSDs during the wet periods. The mean DSDs are concave downward during both the spells. The mean concentration of smaller drops (< 0.9 mm) is higher, and the mean concentration of medium and larger drops is lower in dry periods. An
increased concentration in smaller drops and a decrease in medium and larger drops concentration is found in the dry spells compared to the seasonal mean concentration. This indicates the collision and breakup processes, as described by Rosenfeld and Ulbrich (2003) and Konwar et al. (2014). In contrast,



low concentrations of smaller drops and an increase in number concentration of drops above 0.9 mm diameter are observed in the wet spells.

To study the differences in DSD during the wet and dry spells with rain rate, the distribution of $N(D)$ is compared at different rain rates, as shown in Figure 6. Here $N(D)$ is plotted on a logarithmic scale. It is evident from this figure that significant differences exist in $N(D)$ from wet to dry spells. The contours are shifted to higher rain rates and higher diameters in the wet spells. It indicates that the mid-size drops in the range 1-2 mm are higher in wet spells than in dry spells for the same rain rate. This

result is more pronounced in lower rain rates below 10 mm $hr^{-1}$. Further, the concentration of raindrops in the range 1-2 mm increases as the rain rate increases between 5-15 mm $hr^{-1}$ during the wet periods. At higher rain rates (above 10 mm $hr^{-1}$), the smaller and mid-size drops are higher in the wet spells than in the dry periods. However, this difference decreases gradually as rain rate increases. At above 30 mm $hr^{-1}$, both the periods show a similar distribution of $N(D)$ (not shown in the figure). However, in the

larger drop diameters above 4.5 mm, the concentration is higher in the wet spells compared to the dry periods in all rain rate intervals (not shown in the figure).

        Figure 7 presents the histograms of DSD parameters, $D_m$, $\log_{10}(N_w)$, $\Lambda$, and $\mu$ during the wet and dry spells. The histograms of $D_m$ are positively skewed during both wet and dry periods (Figure 7a). The distribution of $D_m$ is broader in the dry spells. The $D_m$ value varies from 0.42 to 4.8 mm, with the

maximum occurrence at ~1.2 mm during the wet periods, whereas it ranges from 0.4 to 5 mm, with the maximum appearance at ~0.8 mm during the dry spells. For $D_m$ values < 1 mm, the distribution for the dry spells is higher than for the wet spells. This finding indicates the predominance of smaller drops during the dry spells. The mean value of $D_m$, along with the standard deviation and skewness, are



provided in Table 2. The mean value of $D_m$ is 1.3 mm, and its standard deviation is 0.38 during the wet

spells, whereas the mean $D_m$ is 0.9 mm, and its standard deviation is 0.37 during the dry spells. A

relatively large number of small drops reduce the $D_m$ value in the dry spells, while the presence of fewer

smaller drops and relatively more mid-size drops increases the $D_m$ value in the wet periods. The

histograms of $\log_{10}(N_w)$ are negatively skewed during both wet and dry spells (Figure 7b). The $\log_{10}(N_w)$

shows an inverse relation with $D_m$ and is varied from 0.52 to 5.11 during the wet spells and 0.50 to 5.43

during the dry periods. The histogram of $\log_{10}(N_w)$ peak at 3.9 during the wet periods. The histograms

of $\log_{10}(N_w)$ shows a bimodal distribution during the dry spells. This bimodal distribution of $\log_{10}(N_w)$

peaks at 3.9 and 5. This finding is consistent with the results of Utsav et al. (2019). They analyzed the 0

dBZ echo top heights, which represent the cloud top heights during wet and dry spells. They observed a

bi-modal distribution in 0 dBZ echo top height, which peaks at 3 km and 6.5 km during the dry periods.

The large value of standard deviation indicates the large variations in $D_m$ and $N_w$ during both wet and

dry periods. The histograms of slope parameter ($\Lambda$) and shape parameter ($\mu$) are shown in Figure 7(c)-

(d). The slope parameter $\Lambda$ represents the truncation of the DSD tail with the raindrop diameter. If the $\Lambda$

values are small, the DSD tail is extended to the larger diameter and vice-versa. The shape parameter $\mu$

indicates the breadth of DSD. The positive (negative) values of $\mu$ indicate the concave downward

(upward) shape for the DSD. The zero value of $\mu$ represents the exponential shape for DSD (Ulbrich,

1983). The histogram of $\Lambda$ shows positive values during both wet and dry spells. The occurrence of $\Lambda$ is

higher below 10 mm$^{-1}$ during the wet periods, indicating the broader spectrum of raindrops, whereas it

is distributed up to 20 mm$^{-1}$ during the dry spells. The extension of $\Lambda$ towards higher values represents

the higher occurrence of smaller drops during both periods. Relatively smaller values of $\Lambda$ and $N_w$





during the wet spells indicates that the tail of the DSD extends to large raindrop sizes. The histogram of

$\mu$ shows positive values during both wet and dry spells indicating the concave downward shape of DSD

during both the periods.

Numerous studies have been carried out to understand the DSDs during different storms and

within a storm (Dolman et al., 2011; Munchak et al., 2012; Friedrich et al., 2013; Thompson et al.,

2015; Dolan et al., 2018). These studies showed the combined dynamical (stratiform and convective)

and microphysical processes occurring in the storms. Therefore, to understand the effect of dynamical

processes on different DSD characteristics during the wet and dry spells, the precipitation events are

classified into stratiform and convective types based on Bringi et al. (2003). To classify precipitation

into stratiform and convective types, Bringi et al. (2003) considered 5 consecutive 2 min DSD samples.

However, in the present study, 10 consecutive 1 min DSD samples are considered to classify the rainfall

as stratiform and convective. If the mean rain rate of 10 successive DSD samples is greater than 0.5 mm

$hr^{-1}$, and if the standard deviation of 10 consecutive DSD samples is less than 1.5 mm $hr^{-1}$, then the

precipitation is classified as stratiform; otherwise, it is classified as convective.

Figure 8 presents the histograms of $D_m$, $\log_{10}(N_w)$, $\Lambda$, and $\mu$ during stratiform rain events in wet

and dry spells. The mean, standard deviation, and skewness of these parameters are provided in Table 3.

The histograms of $D_m$ (Figure 8a) are positively skewed during stratiform rain events in both the spells.

The histogram of $D_m$ is broader in dry spells. It varies between 0.38 and 2.77 mm with maximum

occurrence near 0.42-0.58 mm during stratiform rain in the dry spells. The distribution of $D_m$ shows a

higher frequency below 0.6 mm in the dry spells. This finding indicates that the presence of more

number of smaller raindrops in stratiform rain of dry spells. The value of $D_m$ varies from 0.42 to 2.48



mm with a maximum near 1-1.4 mm during stratiform rain in the wet periods. The distribution of $D_m$ is higher in the wet spells above 1 mm, indicating the dominance of medium size and/or larger drops in stratiform rain of wet periods. The histogram of $\log_{10}(N_w)$ (Figure 8b) is positively skewed in stratiform rain in the wet spells and negatively skewed in stratiform rain in the dry periods. The distribution is narrower in the wet periods and broader in the dry spells. The distribution peaks between 3-3.6 during the wet spells, whereas it peaks at 5 during the dry spells. The distribution of $\Lambda$ (Figure 8c) is broader in the stratiform rain events during both wet and dry periods. The distribution varies from 1.2 mm$^{-1}$ to 52 mm$^{-1}$ with a mode at 10 mm$^{-1}$ in the stratiform rain of wet spells. This result further supports the presence of mid-size drops during the wet periods. The distribution of $\Lambda$ shows higher occurrences above 15 mm$^{-1}$ during the dry spells, indicating the truncation of DSD at relatively smaller drop diameters. The histograms of $\mu$ (Figure 8d) show a concave downward shape for DSDs during stratiform rain events in both wet and dry spells.

Figure 9 shows the distribution of $D_m$, $\log_{10}(N_w)$, $\Lambda$, and $\mu$ during convective rain events in wet and dry spells. The histograms of $D_m$ are positively skewed in convective rain during both wet and dry spells (Figure 9a). In convective rain, the distribution of $D_m$ is broader in wet spells. It can be seen that the presence of small drops is higher in the dry spells even in convective rain also. The distribution of $\log_{10}(N_w)$ shows an inverse relation with $D_m$ in convective rain (Figure 9b). The $\log_{10}(N_w)$ is negatively skewed in the wet spells, whereas it is positively skewed in the dry spells. The distribution of $\Lambda$ (Figure 9c) indicates the presence of larger drops in convective rain compared to stratiform rain in both wet and dry spells. The histograms of $\mu$ (Figure 9d) show the concave downward shape of DSDs in convective



rain of both wet and dry spells. The mean, standard deviation, and skewness of these parameters are provided in Table 4.

Several points can be noted from the above discussion:

*a*. The maximum value for mean $D_m$ and the largest standard deviation is found for convective rain in wet spells.

*b*. The maximum value for $\log_{10}(N_w)$ and higher standard deviation are observed during stratiform rain in dry spells.

*c*. A considerable difference is found in the histograms of $D_m$ and $\log_{10}(Nw)$ during the stratiform rain in dry and wet periods. However, this difference is small in convective rain.

*d*. In histograms of $\Lambda$ and $\mu$, the distinct differences exist in stratiform rain during wet and dry spells.

The above results indicate that the rainfall over WGs is associated with warm rain processes during both wet and dry spells. The microphysical processes in warm rain include rain evaporation, accretion of cloud water by raindrops and rain sedimentation (Zhang et al., 2008). Giangrande et al. (2017) observed the predominance of larger cloud droplets in warm clouds during the wet spells over Amazon. Similarly, Machado et al. (2018) showed that the larger $D_m$ values are associated with the mixed-phase clouds during the dry periods over Amazon. Recently, Utsav et al. (2019) showed that the presence of cumulus congestus is higher during the wet spells, and shallow clouds are dominant during the dry periods. Thus, the larger values of $D_m$ may be due to the presence of cumulus congestus during the wet spells. To understand the dynamical mechanisms leading to different microphysical processes during wet and dry periods, we have analyzed temperature and specific humidity for monsoon seasons during 2012-2015 over WGs. Figure 10 shows the mean specific humidity (kg kg$^{-1}$) and temperature



anomalies (K) at 700 hPa derived from the ERA-Interim reanalysis dataset. In this plot, the colour bar represents the mean specific humidity, and the contours represent the temperature anomalies. This level is chosen, as the temperature anomaly and the availability of moisture at this level aid the growth of active convection. It is observed that the temperature is cooler over the west coast of India (including the study region) in the wet spells compared to that in the dry periods. Further, the mean specific humidity is higher over WGs during the wet periods. The thermal gradient between WGs and surrounding regions and the availability of more moisture favours the growth of active convection in the wet spells. It is known that the vertical velocity during the wet periods is stronger compared to the dry spells (Uma et al., 2012). The strong updrafts aid the growth of cloud liquid water particles and thereby increase the size of the drops. Whereas, positive temperature anomalies in the dry spell can lead to the evaporation of raindrops, which subsequently can break the drops, thereby leading to lesser diameter drops in the dry spell.

The diurnal variation in mean rain rate during wet and dry spells is shown in Figure 11. The mean rain rate is higher during wet periods throughout the day. The relatively lower rain rates are due to the presence of a higher concentration of smaller drops during the dry spells. The diurnal variation in rain rate shows bi-modal distribution during both wet and dry spells. The primary maximum is in the afternoon hours and the secondary maximum present during morning hours. The raindrop concentration increases monotonically (Fig. 4), with an increase in rain rate for all the drop sizes during the dry spells. This finding indicates that the increase in rain rate is responsible for the rise in both concentration and raindrop size during the dry spells. However, in the wet periods, the concentration of smaller drops is constant throughout the day, and the increase in rain rate is due to the rise in concentration and size of





mid-size raindrops. This further indicates that the collision and coalescence processes as well as deposition of water vapour on to the cloud drops, which are responsible for the increase in the

concentration (afternoon and early morning hours) of mid-size raindrops during the wet spells. In addition, the raindrop diameter depends on the rain rate, which varies between wet and dry spells. The distribution of $D_m$ during wet and dry spells at different rain rates are shown in Figure 12. For lower rain rates (below 10 mm hr$^{-1}$), the raindrops falling from the cloud tops can grow by deposition of water vapour and accretion of cloud water during the wet spells. This result in larger $D_m$ values during the wet

spells compared to dry spells. At higher rain rates (above 20 mm hr$^{-1}$), the $D_m$ distribution remains the same during both the spells. This is due to the equilibrium of DSD by the collision, coalescence, and breakup mechanisms, as described in Hu and Srivastava (1995) and Atlas and Ulbrich (2000). The above analysis indicates that the dynamical mechanisms are different during wet and dry spells, resulting in different DSD characteristics.

*5.2. Implications of DSD during wet and dry spells: μ-Λ relation*

The gamma distribution function has been widely used in the microphysical parameterization schemes in the atmospheric models to describe various DSDs. However, $\mu$ is often considered to be constant. Milbrandt and Yau (2005) found that $\mu$ plays a vital role in determining sedimentation and microphysical growth rates. In this context, the microphysical properties of clouds and precipitation are

sensitive to variations in $\mu$. Several researchers showed that the value of $\mu$ varies during the precipitation (Ulbrich, 1983; Ulbrich and Atlas, 1998; Testud et al., 2001; Zhang et al., 2001; Islam et al., 2012). Zhang et al. (2003) proposed an empirical $\mu$-$\Lambda$ relationship using 2DVD data collected in Florida. They examined the $\mu$-$\Lambda$ relation with different types of precipitation. These $\mu$-$\Lambda$ relations are





useful in reducing the bias in rain parameters from remote sensing measurements (Zhang et al., 2003).

Recent studies have demonstrated the variability in $\mu$-$\Lambda$ relation in different types of rain and at various

geographical locations (Chang et al., 2009; Kumar et al., 2011; Wen et al., 2016). Hence, it is necessary

to derive different $\mu$-$\Lambda$ relations based on local DSD observations, in particular, over the WGs.

In the present study, an empirical $\mu$-$\Lambda$ relationship is derived for both wet and dry spells. To

minimize the sampling errors, the DSDs with a rainfall rate of less than 5 mm hr$^{-1}$ are excluded. In

addition, the total drop counts above 1000 are only considered in the analysis, as proposed by Zhang et

al. (2003). Figure 13 shows the $\mu$-$\Lambda$ relation for wet and dry spells, and the corresponding polynomial

least-square fits are shown as solid lines. The fitted $\mu$-$\Lambda$ relations for wet and dry spells are given as

follows:

Wet spell: $$\Lambda = 0.0359\mu^2 + 0.802\mu + 2.22 \qquad (14)$$

Dry spell: $$\Lambda = 0.0138\mu^2 + 1.151\mu + 1.198 \qquad (15)$$

Similar behaviour is observed for both wet and dry spells, the smaller the value of $\Lambda$ (higher rain

rates), smaller is the value of $\mu$. Thus, the DSDs tend to be more concave downwards with the increase

in rainfall intensity. This finding suggests a higher fraction of small and mid-size drops and a lower

fraction of larger drops, reflecting less evaporation of smaller drops and more drop breakup processes.

However, the fitted $\mu$-$\Lambda$ relation exhibits a large difference for wet and dry spells. Comparing Eq. (14)

and (15), one can observe that the coefficient of the linear term is smaller in wet spells than that of dry

spells. Hence, for a given value of $\mu$, the dry spells have a higher value of $\Lambda$ compared to the wet spells.

Further, the $D_m$ value is higher during wet spells compared to dry spell for the given rainfall rate due to

different microphysical mechanisms as discussed above (Fig. 12). This leads to higher $\mu$ values in wet





spells compared to dry spells. This result suggests that different microphysical mechanisms during wet

and dry spells lead to different $\mu$-$\Lambda$ relations. Hence, it is apparent that the single $\mu$-$\Lambda$ relation cannot

reliably represent the observed phenomenon during different phases of the monsoon.

Comparing the $\mu$-$\Lambda$ relations in this study with that obtained from Zhang et al. (2003), the $\mu$-$\Lambda$

relationship of the dry spell has a smaller slope. These differences reveal that the DSD during dry spell

have lower values of $D_m$. This indicates that the underlying microphysical processes in the orographic

precipitating systems are different from those observed over Florida in 1998 summer. Further, the $\mu$-$\Lambda$

relationships are derived for convective and stratiform rain for the JWD measurements and are provided

in Figure 14. The least-square polynomial fit for convective and stratiform rain is as follows:

Convective rain: $\qquad$ $\Lambda = 0.0069\mu^2 + 0.576\mu + 2.42$ $\qquad$ (16)

Stratiform rain: $\qquad$ $\Lambda = 0.0022\mu^2 + 0.933\mu + 1.86$ $\qquad$ (17)

It is observed that the coefficients of the squared and linear term of convective precipitation are

smaller than those given by Zhang et al. (2003). Hence, for a given value of $\mu$, the convective

precipitation in the present study gives lower values of $\Lambda$ than that for the convective precipitation from

Zhang et al. (2003).

Seela et al. (2018) fitted $\mu$-$\Lambda$ relations for summer and winter rainfall over North Taiwan. Chen

et al. (2017) have derived an empirical $\mu$-$\Lambda$ relation over Tibetan Plateau. Cao et al. (2008) analyzed the

$\mu$-$\Lambda$ relations over Oklahoma. Different $\mu$-$\Lambda$ relations are derived for different weather systems over

North Taiwan (Chu and Su 2008). The $\mu$-$\Lambda$ relationship obtained in the present study differs from Zhang

et al. (2003), Chu and Su (2008), and Seela et al. (2018). The differences in the $\mu$-$\Lambda$ relations could be

attributed to different geographical locations, different microphysical processes, different rainfall rates,



and different types of instruments. To explore the plausible effect of rainfall rate, the $\mu$-$\Lambda$ relations are compared with the previous studies for rain rates below 5 mm hr$^{-1}$, and above 5 mm hr$^{-1}$ (figure not shown). It is observed that, when the rain rates are below 5 mm hr$^{-1}$, the shape parameter shows bimodal distribution (above $\mu$=10), especially in the wet spells. In this rain rate region, the first distribution (with

lower $\mu$ values) is comparable with Chu and Su (2008), and Zhang et al. (2003), whereas the other distribution (with high $\mu$ values) is comparable with Seela et al. (2018). Chu and Su (2008) derived the $\mu$-$\Lambda$ relations for rain rates above 1 mm hr$^{-1}$, as well as rain rates below 5 mm hr$^{-1}$. Hence, the observed differences in $\mu$-$\Lambda$ relation with Chu and Su (2008) could be attributed to the difference in the rain rates. The second distribution is similar to that observed in the rain rates above 5 mm hr$^{-1}$. The slope of the $\mu$-

$\Lambda$ relation is higher compared to Chu and Su (2008), and Zhang et al. (2003) in the rain rates above 5 mm hr$^{-1}$. This result indicates that the wet and dry spells have higher $\mu$ values compared to the previous studies for the same $\Lambda$ values. This represents that, the underlying microphysical processes are different over the complex orographic region, WGs. It can be observed that the $D_m$ values in the present study are higher compared to the previous studies (e.g., Seela et al., 2018). The different $D_m$ distributions lead to

different $\mu$ values as (Ulbrich, 1983):

$$\Lambda D_m = 4 + \mu \qquad (18)$$

Thus, the relatively higher values of $D_m$ could contribute to higher values of $\mu$ for the same $\Lambda$ values in the present study. Hence, the differences in the $\mu$-$\Lambda$ relations with previous studies may be related to different microphysical processes (such as collision-coalescence, breakup, etc.) occurring in

the rainfall over WGs. In addition, Zhang et al. (2003), Chu and Su (2008) used the 2DVD measurements, whereas, in the present study, JWD data are utilized. The different instruments can have



different sensitivities, which can also affect $\mu$-$\Lambda$ relations. The $\mu$-$\Lambda$ relationships derived for the present study are compared with the other orographic precipitations and are provided in Table 5. It is clear that $\mu$-$\Lambda$ relations vary in different types of rainfall and climatic regimes.


## 6. Summary

The raindrop spectra measured by JWD are analyzed to understand the DSD variations during wet and dry spells of the ISM over the WGs. Observational results indicate that the mean DSDs are considerably different during wet and dry periods. In addition, the DSD variability is studied with

stratiform and convective rain during wet and dry spells. Key findings are listed below:

i.     A high concentration of smaller drops is always present in the WGs region, indicating the dominance of shallow convection.

ii.    The DSD over WGs shows distinct diurnal features. The diurnal variation shows that the concentration of smaller drops is higher in dry spells, while the concentration of mid-size drops

is higher in wet spells throughout the day.

iii.   The dry spells exhibit a strong diurnal cycle with double-peak during late afternoon and night time in both smaller and mid-size drops. Whereas, this diurnal cycle is weak for smaller drops in wet spells.

iv.    The higher concentration of mid-size and larger drops is observed in wet spells compared to dry

spells. The thermal gradient between WGs and surrounding regions, higher availability of water vapour, and strong vertical winds favours the formation of cumulus congestus, which are responsible for the presence of medium size/larger drops during wet spells.

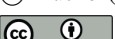


v. The DSDs over WGs are characterized by small $D_m$, and large $N_w$. The $N_w$ shows a bi-modal distribution during dry spells. This bimodality is weak in the wet spells.

vi. The distribution of $\Lambda$ shows the dominance of small drops in dry spells and the dominance of mid-size drops in wet spells. The distribution of $\mu$ represents the concave downward shape of DSDs for both wet and dry spells.

vii. An empirical relation is derived between $\mu$ and $\Lambda$ during wet and dry spells. The fitted $\mu$-$\Lambda$ relationship for both spells exhibits a significant difference between them. The different

microphysical mechanisms lead to different $\mu$-$\Lambda$ relations during wet and dry spells.

viii. A considerable difference in raindrop size distribution is observed in the stratiform rain of wet and dry spells. Higher amounts of smaller drops are evident in both stratiform and convective rain of dry spells compared to wet spells.

It is evident from this study that, even though the warm rain is predominant, the dynamical

mechanisms underlying the microphysical processes are different, which causes the difference in observed DSD characteristics during wet and dry spells. The distinct features of DSD during the wet and dry spells of the ISM over WGs are summarized in Figure 15.

**Author contributions:**

UVMK and SKD designed, analyzed, and prepared the manuscript. SKD, UVMK, and UB proposed the methodology. GSE, SMD, and GP contributed with discussion to the manuscript.

**Acknowledgements:**





The authors are thankful to the Director, IITM, for his support. The authors would like to
acknowledge the technical/administrative staff of the High Altitude Cloud Physics Laboratory
(HAPCL), Mahabaleshwar, for maintaining disdrometer. The authors acknowledge the India
Meteorological Department (IMD) for the provision of the rainfall dataset. The authors also
acknowledge the JAXA, JAPAN, and NASA, USA, for the provision of the GPM data
(https://pmm.nasa.gov/data-access/downloads/gpm). The authors would like to acknowledge the
European Centre for Medium-Range Weather Forecasts (ECMWF) for the provision of the ERA-
Interim dataset. The disdrometer data are archived at IITM and are available with the corresponding
author (skd_ncu@yahoo.com) for research collaboration. The manuscript benefitted from comments
and suggestions provided by the Editor and the anonymous reviewers.



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





**Table Captions:**

**Table 1:** Total number of wet and dry days during the monsoon seasons (June-September) of 2012 - 2015.

**Table 2:** Mean, Standard deviation, and Skewness of the DSD parameters in wet and dry spells.

**Table 3:** Mean, Standard deviation, and Skewness of the DSD parameters in stratiform rain during wet and dry spells.

**Table 4:** Mean, Standard deviation, and Skewness of the DSD parameters in convective rain during wet and dry spells.

**Table 5:** Comparison of $\mu$-$\Lambda$ relations derived in the present study with the orographic precipitation on

other parts of the globe.





**Figure Captions:**

**Fig.1:** Scatter plot of daily accumulated rainfall between rain gauge and JWD. The solid grey line indicates the linear regression.

**Fig 2:** The standardized rainfall anomaly for the year (a) 2012, (b) 2013, (c) 2014, and (d) 2015 during the period June-September. The dashed line marked for 0.5 (+ve X-axis) and -0.5 (-ve X-axis) rainfall anomaly.

**Fig 3:** Box and whisker plot of $D_m$ distributions over the ocean, windward (HACPL), and leeward side of the mountain obtained from GPM measurements. Box represents the data between first and third quartiles, and the whiskers show the data from 12.5 and 87.5 percentiles. The horizontal line within the box represents the median value of the distribution.

**Fig 4:** Diurnal variation in raindrop concentration during wet and dry spells for (a) smaller drops (< 1mm) and (b) mid-size drops (1-4 mm). The concentration of raindrops within each hour is normalized with the total concentration of raindrops in the respective spells (wet or dry). The black line represents wet spells, and the red line represents dry spells.

**Fig 5:** Average DSDs during wet and dry spells.

**Fig 6:** The variation in $N(D)$ as a function of $D$ at different $R$ for (a) wet and (b) dry spells.

**Fig 7:** Histograms of $D_m$, $\log_{10}(N_w)$, $\Lambda$ and $\mu$ during wet and dry spells. The black line represents wet spells, and the red line represents dry spells.

**Fig 8:** Histograms of $D_m$, $\log_{10}(N_w)$, $\Lambda$ and $\mu$ in stratiform rain during wet and dry spells. The black line represents wet spells, and the red line represents dry spells.





**Fig 9:** Histograms of $D_m$, $\log_{10}(N_w)$, $\Lambda$ and $\mu$ in convective rain during wet and dry spells. The black line represents wet spells, and the red line represents dry spells.

**Fig 10:** Spatial distribution of mean specific humidity (kg kg$^{-1}$), and temperature anomalies (K) at 700 hPa during (a) wet and (b) dry spells of the monsoon seasons of 2012-2015. The colour bar represents the specific humidity, and contours represent temperature anomalies. The positive anomaly represents heating, and negative anomaly represents cooling. The black dot represents the observational site.

**Fig 11:** Diurnal variation of mean rain rate (mm hr$^{-1}$) during wet and dry spells.

**Fig 12:** Distribution of $D_m$ at different rain rates during wet and dry spells. The horizontal line within the box represents the median value. The boxes represent data between first and third quartiles, and the whiskers show data from 12.5 to 87.5 percentiles. The black colour represents wet spells, and the red colour represents dry spells.

**Fig 13:** Scatter plots of $\mu$-$\Lambda$ values obtained from gamma DSD for (a) wet and (b) dry spells. The solid line indicates the least square polynomial fit for $\mu$-$\Lambda$ relation.

**Fig 14:** Scatter plots of $\mu$-$\Lambda$ values obtained from gamma DSD for (a) convective and (b) stratiform rain. The solid line indicates the least square polynomial fit for $\mu$-$\Lambda$ relation.

**Fig 15:** Summary of the DSD characteristics during the wet and dry spells in the WGs region.





**Table 1: Total number of wet and dry days during the monsoon seasons (June-September) of 2012 – 2015.**

| Months | Wet (No. of. Days) | Dry (No. of. Days) |
|---|---|---|
| June | 15 | 40 |
| July | 16 | 38 |
| August | 0 | 46 |
| September | 10 | 35 |

**Table 2: Mean, Standard deviation, and Skewness of the DSD parameters in wet and dry spells.**

| | Wet | | | Dry | | |
|---|---|---|---|---|---|---|
| | **Mean** | **Standard deviation** | **Skewness** | **Mean** | **Standard deviation** | **Skewness** |
| $D_m$ | 1.30 | 0.38 | 0.56 | 0.92 | 0.37 | 1.41 |
| $\log_{10}(N_w)$ | 3.62 | 0.51 | -0.52 | 4.46 | 0.68 | -0.23 |
| $\Lambda$ | 15.42 | 10.25 | 1.17 | 22.01 | 12.43 | 0.48 |
| $\mu$ | 14.40 | 9.94 | 1.09 | 17.80 | 11.02 | 0.70 |
| R | 6.62 | 9.75 | 3.19 | 2.79 | 5.02 | 4.59 |





**Table 3: Mean, Standard deviation, and Skewness of the DSD parameters in stratiform rain during wet and dry spells.**

| | Wet spells | | | Dry spells | | |
|---|---|---|---|---|---|---|
| | Mean | Standard deviation | Skewness | Mean | Standard deviation | Skewness |
| $D_m$ | 1.18 | 0.31 | 0.14 | 0.75 | 0.265 | 1.28 |
| $\log_{10}(N_w)$ | 3.52 | 0.56 | 0.19 | 4.39 | 0.68 | -0.69 |
| $\Lambda$ | 17.08 | 10.56 | 0.97 | 26.77 | 12.48 | 0.61 |
| $\mu$ | 15.12 | 10.17 | 1.02 | 20.81 | 10.76 | 0.40 |

**Table 4: Mean, Standard deviation, and Skewness of the DSD parameters in convective rain during wet and dry spells.**

| | Wet spells | | | Dry spells | | |
|---|---|---|---|---|---|---|
| | Mean | Standard deviation | Skewness | Mean | Standard deviation | Skewness |
| $D_m$ | 1.66 | 0.29 | 0.88 | 1.47 | 0.30 | 0.34 |
| $\log_{10}(N_w)$ | 3.86 | 0.23 | -0.54 | 4.01 | 0.29 | 0.19 |
| $\Lambda$ | 10.08 | 5.22 | 1.29 | 13.15 | 7.49 | 1.09 |
| $\mu$ | 11.86 | 6.70 | 0.77 | 14.05 | 8.73 | 1.16 |



**Table 5: Comparison of $\mu$-$\Lambda$ relations derived in the present study with the orographic precipitation on other parts of the globe.**

| Study | Climatic Regime | μ-Λ relation |
|---|---|---|
| **Present study** | **Wet spells over WGs** | $\Lambda = 0.0359\mu^2 + 0.802\mu + 2.22$ |
| **Present study** | **Dry spells over WGs** | $\Lambda = 0.0138\mu^2 + 1.151\mu + 1.198$ |
| **Present study** | **Stratiform precipitation** | $\Lambda = 0.0022\mu^2 + 0.933\mu + 1.86$ |
| **Present study** | **Convective precipitation** | $\Lambda = 0.0069\mu^2 + 0.576\mu + 2.42$ |
| **Seela et al. (2018)** | **Summer season in Taiwan** | $\Lambda = 0.0235\mu^2 + 0.472\mu + 2.394$ |
| **Seela et al. (2018)** | **Winter season in Taiwan** | $\Lambda = -0.0135\mu^2 + 1.006\mu + 3.48$ |
| **Chen et al. (2017)** | **Summer season in Tibetan Plateau** | $\Lambda = -0.0044\mu^2 + 0.764\mu - 0.49$ |
| **Cao et al. (2008)** | **Oklahoma** | $\Lambda = -0.02\mu^2 + 0.902\mu - 1.718$ |
| **Chu and Su (2008)** | **Typhoons in north Taiwan** | $\Lambda = 0.0433\mu^2 + 1.039\mu + 1.477$ |
| **Zhang et al. (2003)** | **Florida** | $\Lambda = 0.0365\mu^2 + 0.735\mu + 1.935$ |



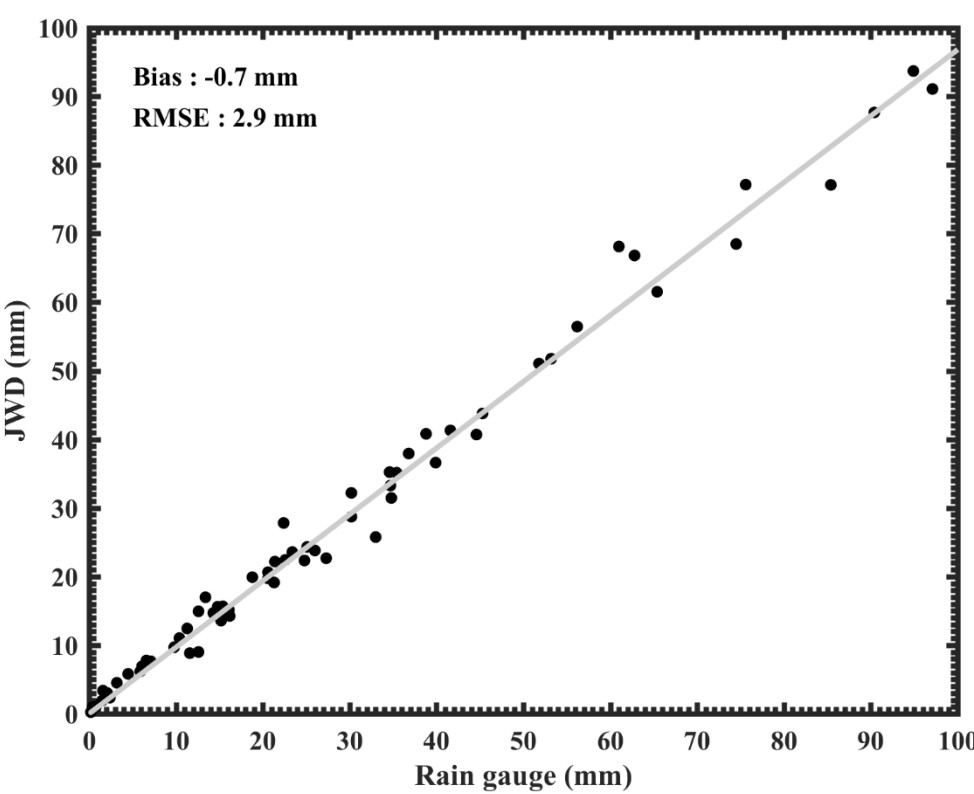

**Fig.1:** Scatter plot of daily accumulated rainfall between rain gauge and JWD. The solid grey line

indicates the linear regression.





**Fig 2:** The standardized rainfall anomaly for the year (a) 2012, (b) 2013, (c) 2014, and (d) 2015 during

the period June-September. The dashed line marked for 0.5 (+ve X-axis) and -0.5 (-ve X-axis)

rainfall anomaly.





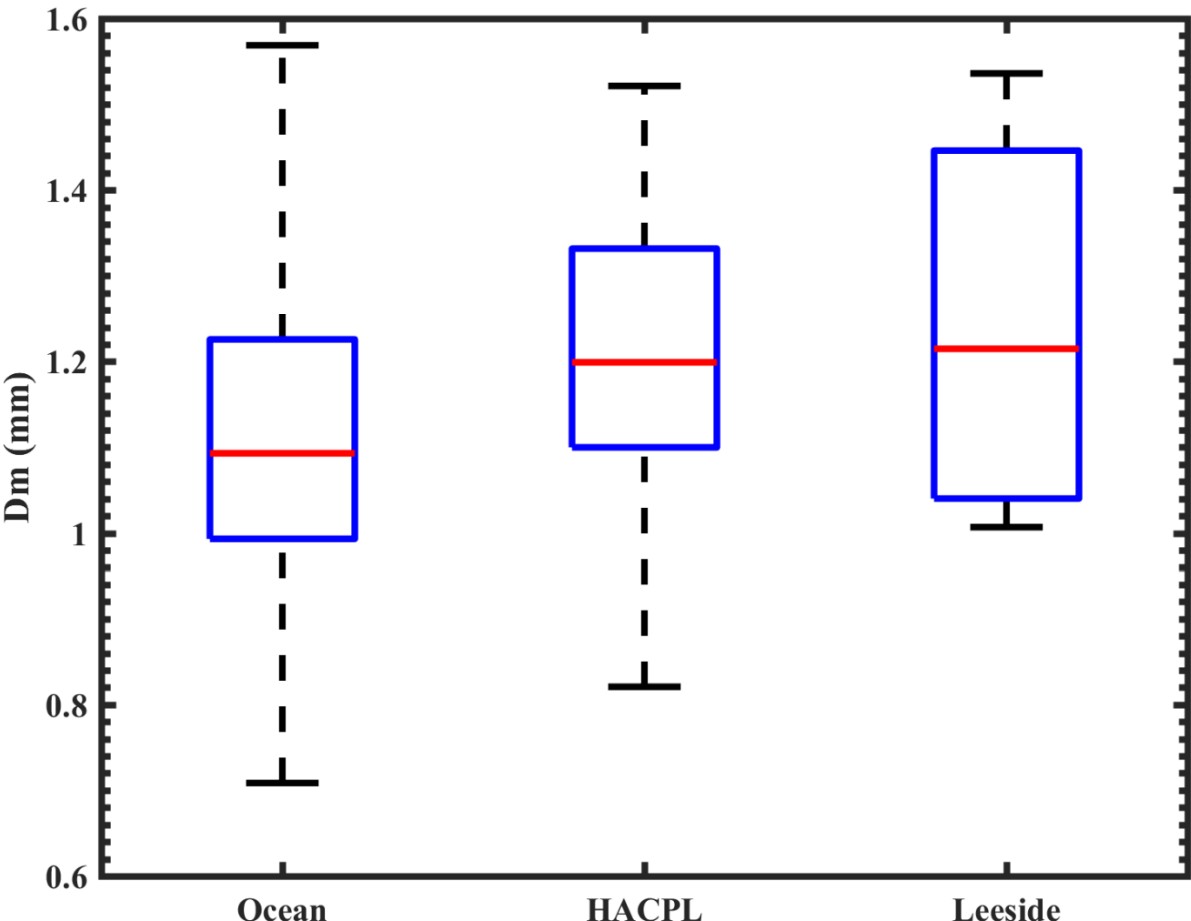

**Fig 3:** Box and whisker plot of $D_m$ distributions over the ocean, windward (HACPL), and leeward side

of the mountain obtained from GPM measurements. Box represents the data between first and

third quartiles, and the whiskers show the data from 12.5 and 87.5 percentiles. The horizontal

line within the box represents the median value of the distribution.





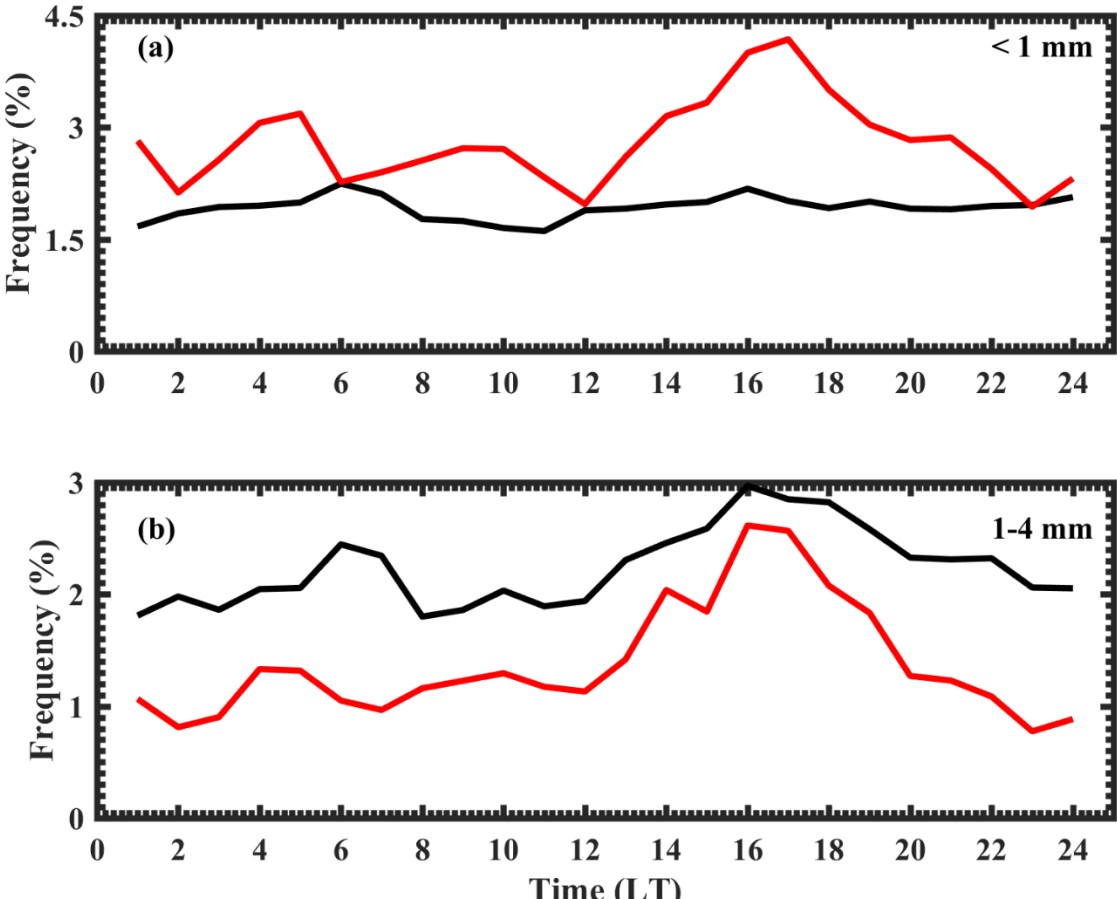

**Fig 4:** Diurnal variation in raindrop concentration during wet and dry spells for (a) smaller drops (< 1mm) and (b) mid-size drops (1-4 mm). The concentration of raindrops within each hour is normalized with the total concentration of raindrops in the respective spells (wet or dry). The black line represents wet spells, and the red line represents dry spells.





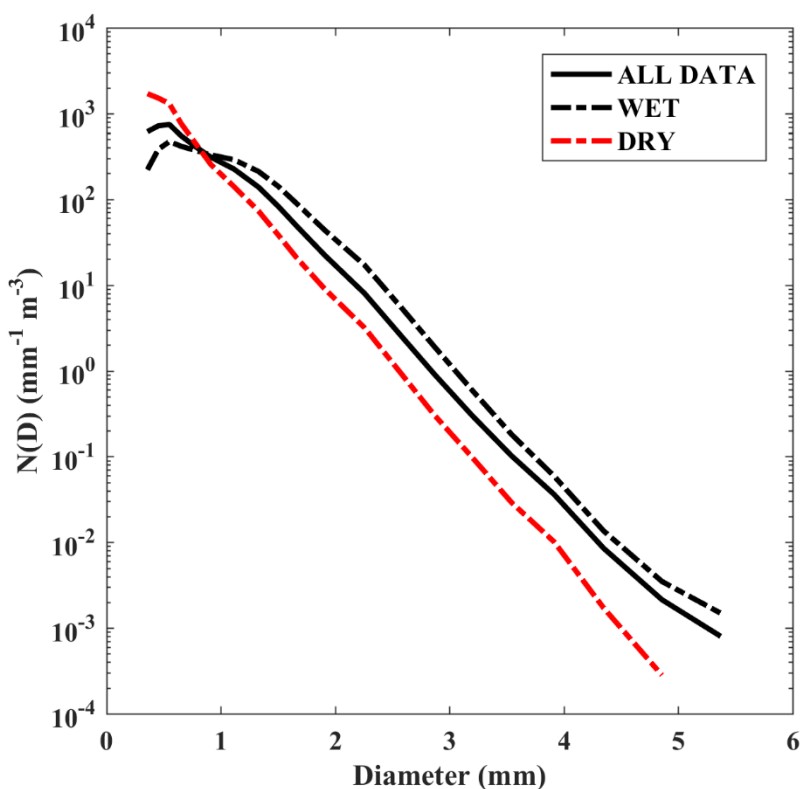

**Fig 5:** Average DSDs during wet and dry spells.


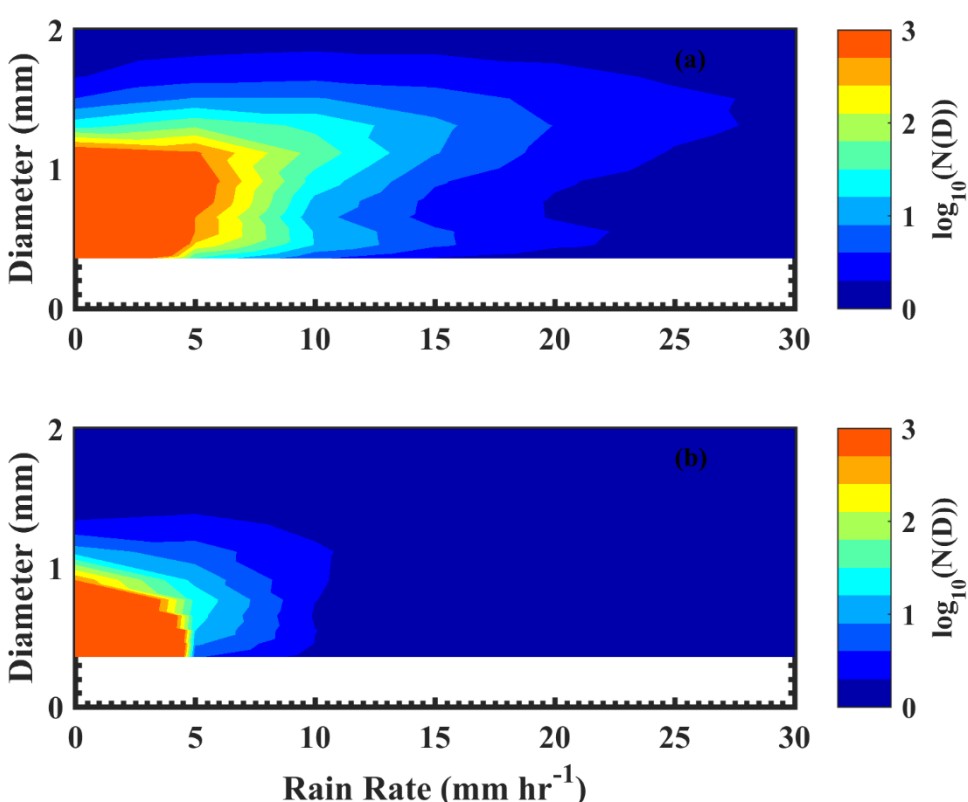

**Fig 6:** The variation in *N(D)* as a function of *D* at different *R* for (a) wet and (b) dry spells.





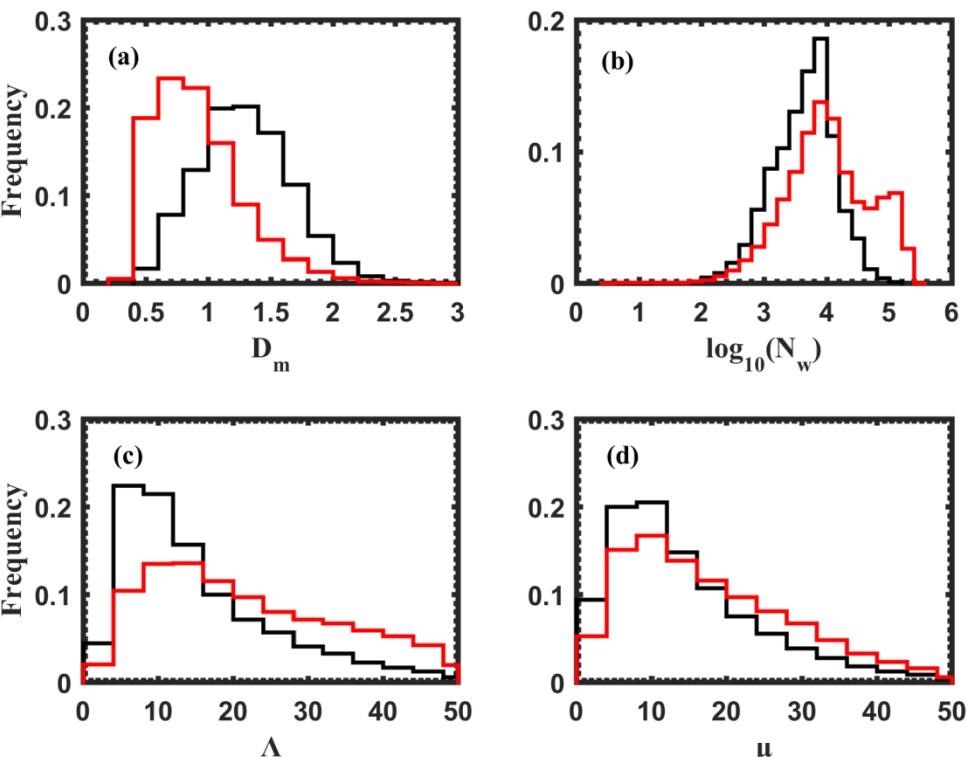

**Fig 7:** Histograms of $D_m$, $\log_{10}(N_w)$, $\Lambda$ and $\mu$ during wet and dry spells. The black line represents wet

spells, and the red line represents dry spells.



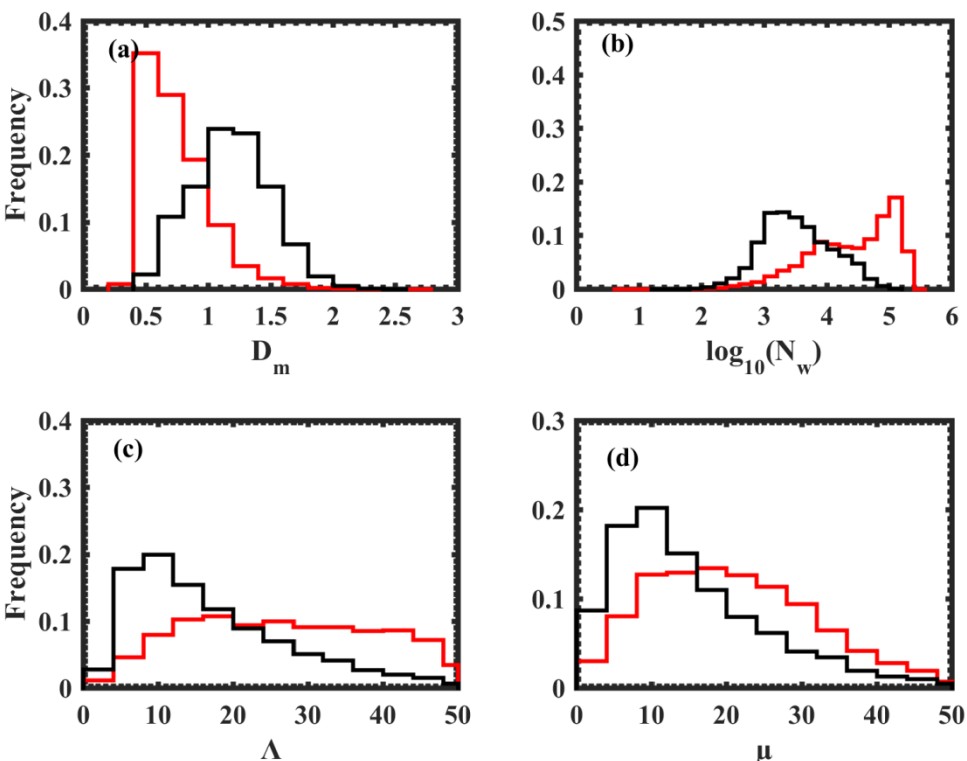

**Fig 8:** Histograms of $D_m$, $\log_{10}(N_w)$, $\Lambda$ and $\mu$ in stratiform rain during wet and dry spells. The black line

represents wet spells, and the red line represents dry spells.




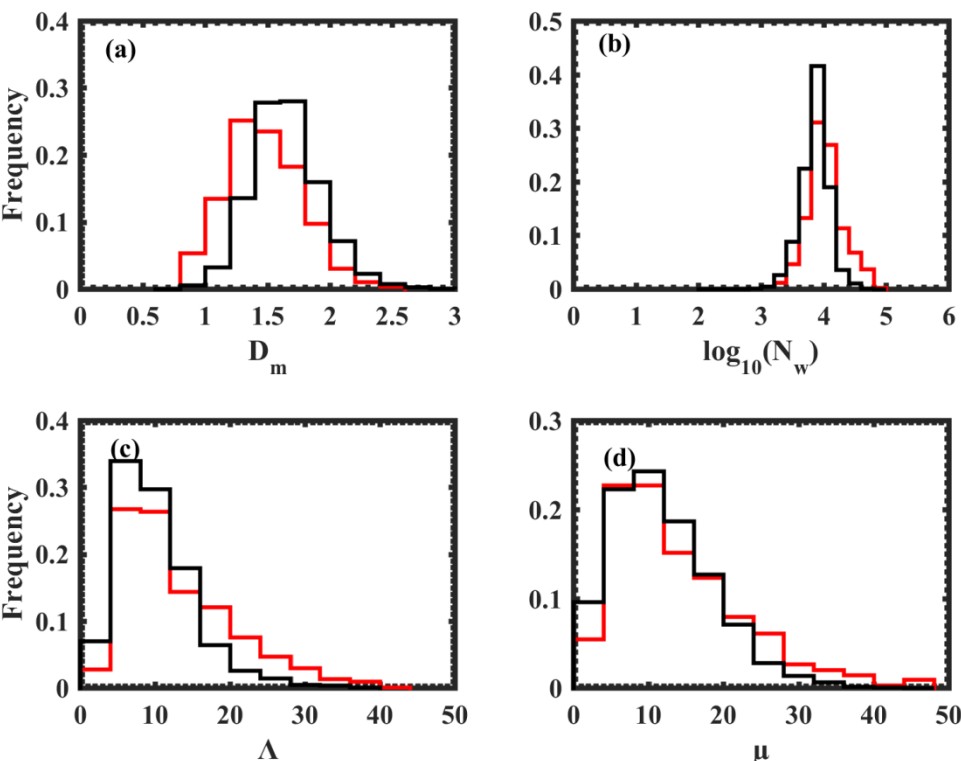

**Fig 9:** Histograms of $D_m$, $\log_{10}(N_w)$, $\Lambda$ and $\mu$ in convective rain during wet and dry spells. The black line represents wet spells, and the red line represents dry spells.

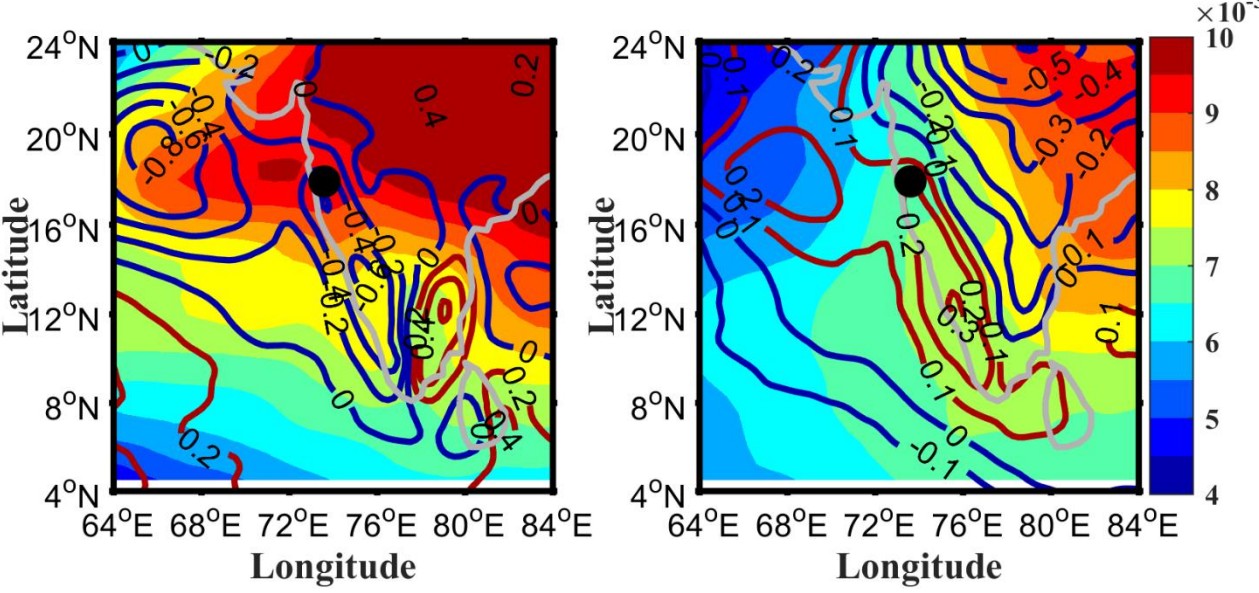

**Fig 10:** Spatial distribution of mean specific humidity (kg kg$^{-1}$), and temperature anomalies (K) at 700

hPa during (a) wet and (b) dry spells of the monsoon seasons of 2012-2015. The colour bar

represents the specific humidity, and contours represent temperature anomalies. The positive

anomaly represents heating, and negative anomaly represents cooling. The black dot represents

the observational site.



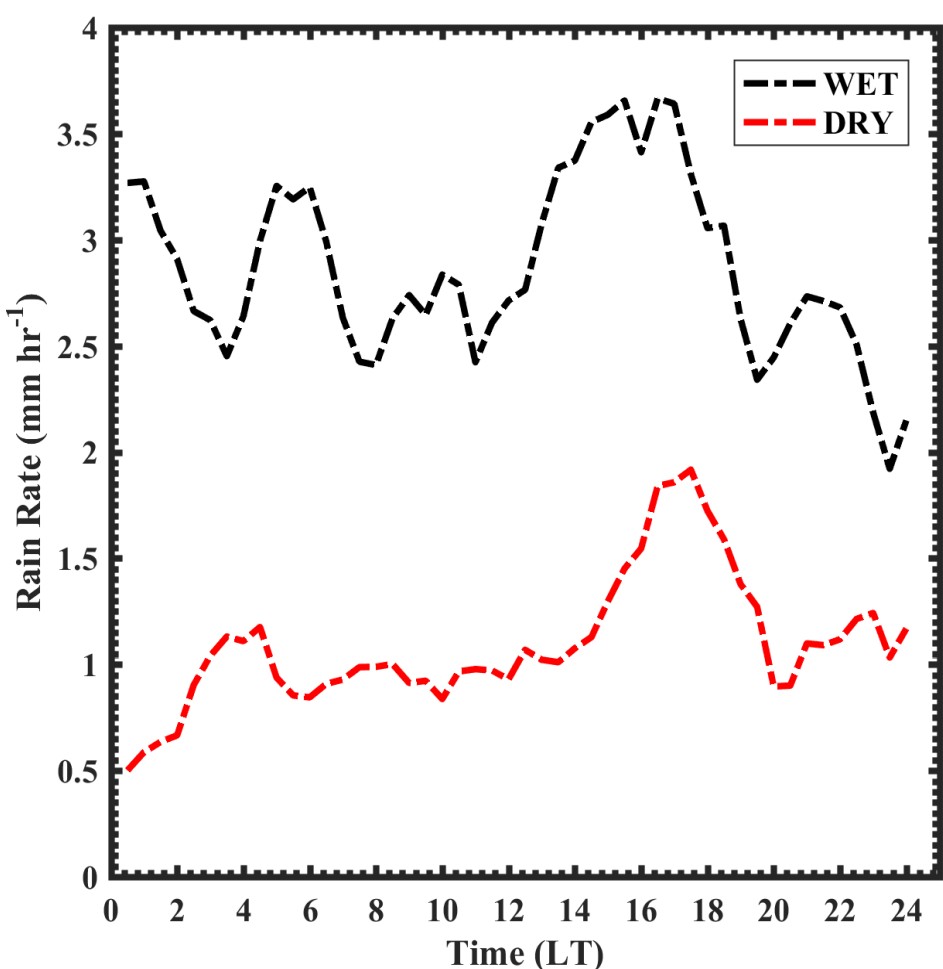

**Fig 11:** Diurnal variation of mean rain rate (mm hr[-1]) during wet and dry spells.



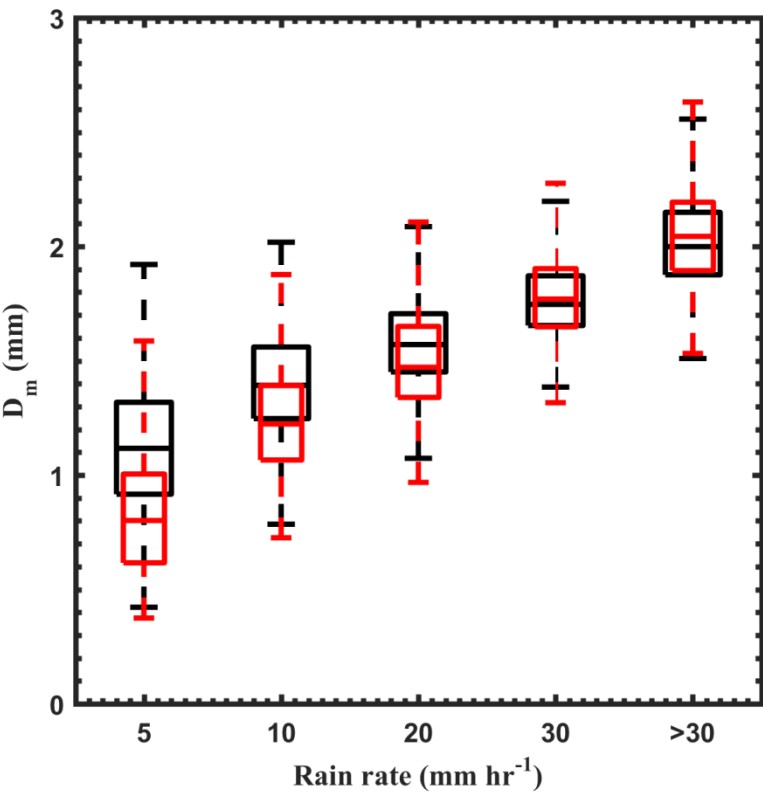

**Fig 12:** Distribution of $D_m$ at different rain rates during wet and dry spells. The horizontal line within

the box represents the median value. The boxes represent data between first and third quartiles,

and the whiskers show data from 12.5 to 87.5 percentiles. The black colour represents wet spells,

and the red colour represents dry spells.





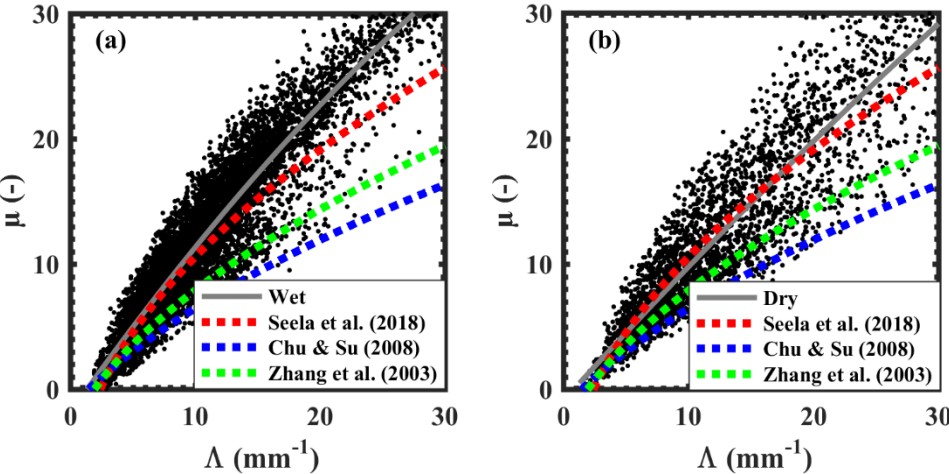

**Fig 13:** Scatter plots of *μ-Λ* values obtained from gamma DSD for (a) wet and (b) dry spells. The solid

line indicates the least square polynomial fit for *μ-Λ* relation.





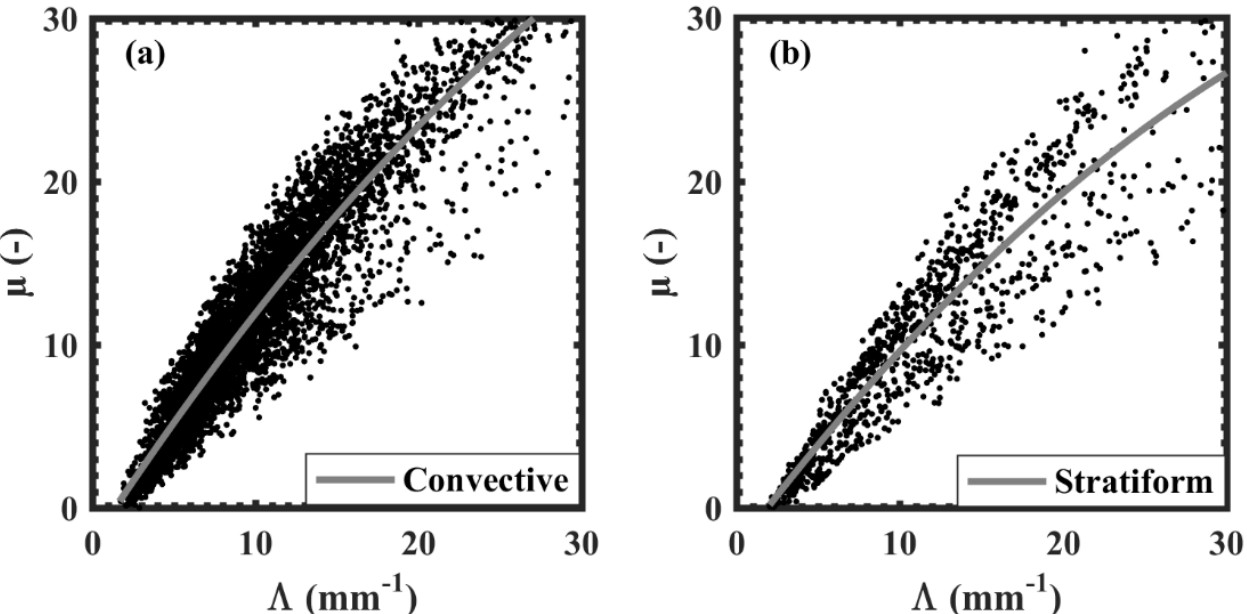

**Fig 14:** Scatter plots of *μ-Λ* values obtained from gamma DSD for (a) convective and (b) stratiform

rain. The solid line indicates the least square polynomial fit for *μ-Λ* relation.





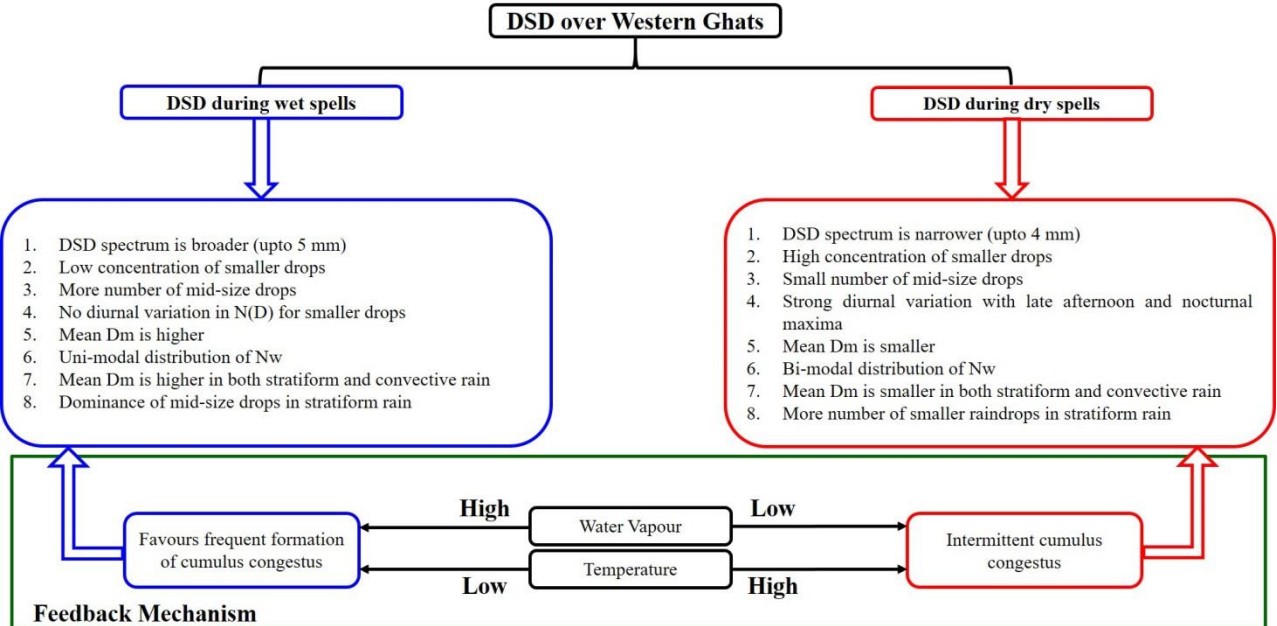

**Fig 15:** Summary of the DSD characteristics during wet and dry spells in the WGs region.