# Peer review of "Statistical characteristics of raindrop size distribution over Western Ghats of India: wet versus dry spells of Indian Summer Monsoon"

_Atmospheric Chemistry and Physics, 2020_

## Referee Comment (RC1) · Anonymous Referee #1 · 29 Dec 2020

Manuscript is well organised and brings out very good results and can be published in this journal. How ever, the paper can be accepted after incorporating appropriately the following points suggested.

Recommendation: Minor revisions needed.

Comments:

1. Being a DSD study and considering its application, authors are suggested to add more references on the studies of DSD for the Indian region in the introduction or at discussion part. This will also clarify clearly the gap in this area of research and add uniqueness to this study. There are many more studies by Harikumar et al., Kiran

[Figure]

Kumar et al., Reddy and Kozhu etc and others for Indian region.

2. Add the reference Sasikumar et al. (2007) in the JESS, which is so important to be referred. Because the rain rate distribution is important and it was found out by them that Weibull distribution fits well for the rain rate occurrence. And say about the presence of low intense rain is more. So, the DSD against such low rain rates are to be looked into while the authors explain the results and at least the readers will keep that important aspect in their mind.

3. Equations 1 to 4 are not to be shown here. There are many papers from India already given these equations in those. You may cite those papers and refer.

4. Altitudinal variation of DSD and what happens as rain falls down to be just mentioned in the study for our region. Below reference will help

R. Harikumar, V. Sasi Kumar and S. Sampath, 'Altitudinal and temporal Evolution of Rain Drop Size Distribution observed over a tropical station using a K-Band Radar', International J. Remote Sensing, 33 (20), 3286-3300, 2012, DOI: 10.1080/01431161.2010.549853.

R. Harikumar, S. Sampath and V. Sasi Kumar, 'An Empirical Model for the Variation of Rain Drop Size Distribution with Rain Rate at a few Locations in Southern India', Adv. in Space Research, 43, 837-844, 2008, DOI: 10.1016/j.asr.2008.11.001.

5. Separation in to startiform and convective is to be explained in detail. It should be connected to the literature. There are many methodologies for that. The studies in this regard in this region to be cited and referred at least in the introduction and to be connected to it. And it should justify the sanctity of the methodology authors applied in this study. Following paper explains that in detail for tropical region/India. R. Harikumar, 'Discernment of near-oceanic precipitating clouds into convective or stratiform based on Z–R model over an Asian monsoon tropical site', Meteorology and Atmospheric Physics, 2019, https://doi.org/10.1007/s00703-019-00696-3 =========

---

## Referee Comment (RC2) · Anonymous Referee #2 · 31 Dec 2020

1. Lines 105-110: The research gaps in this study need to be further clarified. As the DSD statistics have been reported in the previous literatures in the Western Ghats. The authors could deepen their exploration on the mesoscale processes associated with the precipitation dynamics of Indian Summer Monsoon.

2. Line 118-121: Please add a figure to show the topographic map and location of the disdrometers used in this study.

3. Line 190-196: Please explain the selection of temperature and specific humidity at 700 hPa as the data sources of water vapor analysis. Why the wind feature is not considered in the analysis. Also, the spatial resolution of ERA-Interim is 0.25 x 0.25

deg, and the disdrometer data is site-based. How to fill and explain the uncertainty of the spatial resolution between ERA-Interim and in-situ disdrometer.

4. Line 374-378: The precipitation classification algorithm needs to be further clarified as the Bringi et al. (2003) approach has been improved since 2003.

5. Line 564-567: The explanation of feedback mechanism during the wet and dry spells are weak. Although Fig.10 shows the temperature and specific humidity patterns during wet and dry spells, lacks of enough quantitative analysis. Please give more evidences to support the feedback mechanism.

---

## Author Comment (AC1) · 25 Jan 2021

| | |
|---|---|
| **General Comment** | *Manuscript is well organised and brings out very good results and can be published in this journal. However, the paper can be accepted after incorporating appropriately the following points suggested.* *Recommendation: Minor revisions needed.* |
| **Response** | **We are indebted to the reviewer for valuable and thoughtful comments on the manuscript. We much appreciate the reviewer's time and efforts during the evaluation of the manuscript. We went through all the referee comments and suggestions and implemented the same in the revised manuscript. Point-to-point clarifications for the referee's comments and how we have addressed each recommendation is listed below. The manuscript is also altered by considering the other reviewer's comments.** |

**Specific Comments**

| | |
|---|---|
| **Comment#1** | *Being a DSD study and considering its application, authors are suggested to add more references on the studies of DSD for the Indian region in the introduction or at discussion part. This will also clarify clearly the gap in this area of research and add uniqueness to this study. There are many more studies by Harikumar et al., Kiran Kumar et al., Reddy and Kozhu etc and others for Indian region.* |
| **Response** | **Thanks. Previous studies on DSD variability over Indian region are added in the revised manuscript.** |

| | |
|---|---|
| **Comment#2** | *Add the reference Sasikumar et al. (2007) in the JESS, which is so important to be referred. Because the rain rate distribution is important and it was found out by them that Weibull distribution fits well for the rain rate occurrence. And say about the presence of low intense rain is more. So, the DSD against such low rain rates are to be looked into while the authors explain the results and at least the readers will keep that important aspect in their mind.* |
| **Response** | **The reference Sasikumar et al. (2007) is added.** |

| | |
|---|---|
| **Comment#3** | *Equations 1 to 4 are not to be shown here. There are many papers from India already given these equations in those. You may cite those papers and refer.* |
| **Response** | **Reviewer's suggestion is implemented.** |

| | |
|---|---|
| **Comment#4** | *Altitudinal variation of DSD and what happens as rain falls down to be just mentioned in the study for our region. Below reference will help* *R. Harikumar, V. Sasi Kumar and S. Sampath, 'Altitudinal and temporal Evolution of Rain Drop Size Distribution observed over a tropical station using a K Band Radar', International J. Remote Sensing, 33 (20), 3286-3300, 2012, DOI:10.1080/01431161.2010.549853.* *R. Harikumar, S. Sampath and V. Sasi Kumar, 'An Empirical Model for the Variation of Rain Drop Size Distribution with Rain Rate at a few Locations in* |

|  |  |
|---|---|
| | *Southern India', Adv. in Space Research, 43, 837-844, 2008, DOI: 10.1016/j.asr.2008.11.001.* |
| **Response** | **Thanks for the suggestion. Now, the importance of altitudinal variation of DSD is added in the introduction with appropriate references.** |
| | |
| **Comment#5** | *Separation in to startiform and convective is to be explained in detail. It should be connected to the literature. There are many methodologies for that. The studies in this regard in this region to be cited and referred at least in the introduction and to be connected to it. And it should justify the sanctity of the methodology authors applied in this study. Following paper explains that in detail for tropical region/India. R. Harikumar, 'Discernment of near-oceanic precipitating clouds into convective or stratiform based on Z–R model over an Asian monsoon tropical site', Meteorology and Atmospheric Physics, 2019, https://doi.org/10.1007/s00703-019-00696-3* |
| **Response** | **Thanks.** |

**Several rain classification schemes proposed in the literature using different instruments, like, disdrometer, radar, profiler (Bringi et al., 2003; Thompson et al., 2015; Krishna et al., 2016; Das et al., 2017; Dolan et al., 2018; Harikumar et al., 2019). In this study, the rainfall at the ground is classified as stratiform and convective based on Bringi et al. (2003) criterion. Even though several other classification schemes available in the literature, it is the most widely used classification criterion for stratiform and convective rainfall. For instance, several past and recent studies (Marzano et al., 2010; Chen et al., 2013; Tnag et al., 2014; Wen et al., 2016; Suh et al., 2016; Wu and Liu, 2017; Seela et al., 2017; 2018) used Bringi et al. (2003) criterion for the classification of precipitation systems. There are slight differences among different classification schemes, which leads to small differences in the DSD characteristics, and hence the choice of different classification schemes is subjective. To the best of author's knowledge, Bringi et al. (2003) criterion didn't have any limitation/drawbacks for analyzing the DSD spectra in the WGs regions. So the authors strongly believe that Bringi et al. (2003) criterion can effectively be used to classify stratiform and convective rain types. As the present study intends to understand the DSD differences between convective and stratiform (rain which does not come under the convective category) rain systems, we adopted the well-known Bringi et al. (2003) criterion. To classify precipitation into stratiform and convective types, Bringi et al. (2003) considered 5 consecutive 2 min DSD samples. However, in the present study, 10 consecutive 1 min DSD samples are considered to classify the rainfall as stratiform and convective. If the mean rain rate of 10 successive DSD samples is greater than 0.5 mm h$^{-1}$, and if the standard deviation of 10 consecutive DSD samples is less than 1.5 mm h$^{-1}$, then the precipitation is classified as stratiform; otherwise, it is classified as convective.**

**Reference:**

Bringi, V. N., Chandrasekar, V., Hubbert, J., Gorgucci, E., Randeu, W. L., and Schoenhuber M.: Raindrop size distribution in different climatic regimes from disdrometer and dual-polarized radar analysis, J. Atmos. Sci., 60, 354–365, 2003.

Chen, B., Yang, J. and Pu, J.: Statistical Characteristics of Raindrop Size Distribution in the Meiyu Season Observed in Eastern China, J. Meteorol. Soc. Japan, 91(2), 215–227 [online] Available from: https://ci.nii.ac.jp/naid/40019636893/en/, 2013.

Das, S. K., Konwar, M., Chakravarty, K. and Deshpande, S M.: Raindrop size distribution of different cloud types over the Western Ghats using simultaneous measurements from Micro-Rain Radar and disdrometer, Atmos. Res., 186, 72–82, doi:http://dx.doi.org/10.1016/j.atmosres.2016.11.003, 2017.

Dolan, B., Fuchs, B., Rutledge, S. A., Barnes, E. A. and Thompson, E. J.: Primary Modes of Global Drop Size Distributions, J. Atmos. Sci., 75(5), 1453–1476, doi:10.1175/JAS-D-17-0242.1, 2018.

Harikumar, R.: Discernment of near-oceanic precipitating clouds into convective or stratiform based on Z–R model over an Asian monsoon tropical site, Meteorology and Atmospheric Physics, 132:377–390, 2020.

Krishna, U. V. M., Reddy, K. K., Seela, B. K., Shirooka, R., Lin, P.-L. and Pan, C.-J.: Raindrop size distribution of easterly and westerly monsoon precipitation observed over Palau islands in the Western Pacific Ocean, Atmos. Res., 174–175, 41–51, doi:https://doi.org/10.1016/j.atmosres.2016.01.013, 2016.

Marzano, F. S., Cimini, D. and Montopoli, M.: Investigating precipitation microphysics using ground-based microwave remote sensors and disdrometer data, Atmos. Res., 97(4), 583–600, doi:https://doi.org/10.1016/j.atmosres.2010.03.019, 2010.

Sasi Kumar, V., Sampath, S., Vinayak, P. V. S. S. K. and Harikumar, R.: Rainfall intensity characteristics at coastal and high altitude stations in Kerala, J. Earth Syst. Sci., (5), 451–463, 2007.

Seela, B. K., Janapati, J., Lin, P.-L., Reddy, K. K., Shirooka, R. and Wang, P. K.: A Comparison Study of Summer Season Raindrop Size Distribution Between Palau and Taiwan, Two Islands in Western Pacific, J. Geophys. Res. Atmos., 122(21), 11,711-787,805, doi:10.1002/2017JD026816, 2017.

Seela, B. K., Janapati, J., Lin, P.-L., Wang, P. K. and Lee, M.-T.: Raindrop Size Distribution Characteristics of Summer and Winter Season Rainfall Over North Taiwan, J. Geophys. Res. Atmos., 123(20), 11,602-611,624, doi:10.1029/2018JD028307, 2018.

Suh, S.-H., You, C.-H. Lee, D.-I.: Climatological characteristics of raindrop size distributions in Busan, Republic of Korea, Hydrol. Earth Syst. Sci., 20, 193–207, 2016.

Tang, Q., Xiao, H., Guo, C. and Feng, L.: Characteristics of the raindrop size distributions and their retrieved polarimetric radar parameters in northern and southern China, Atmos. Res., 35–136, 59-75, 2014.

Thompson, E. J., Rutledge, S. A., Dolan, B. and Thurai, M.: Drop Size Distributions and Radar Observations of Convective and Stratiform Rain over the Equatorial Indian and West Pacific Oceans, J. Atmos. Sci., 72(11), 4091–4125, doi:10.1175/JAS-D-14-0206.1, 2015.

Wen, L., Zhao, K., Zhang, G., Xue, M., Zhou, B., Liu, S. and Chen, X.: Statistical characteristics of raindrop size distributions observed in East China during the Asian summer monsoon season using 2-D video disdrometer and Micro Rain Radar data, J. Geophys. Res. Atmos., 121(5), 2265–2282, doi:10.1002/2015JD024160, 2016.

Wu, Y. and Liu, L.: Statistical Characteristics of Raindrop Size Distribution in the Tibetan Plateau and Southern China, Adv. Atmos. Sci., 34, 727–736, 2017.

---

## Author Comment (AC2) · 25 Jan 2021

**Comment#1** *Lines 105-110: The research gaps in this study need to be further clarified. As the DSD statistics have been reported in the previous literatures in the Western Ghats. The authors could deepen their exploration on the mesoscale processes associated with the precipitation dynamics of Indian Summer Monsoon.*

**Response** **Several studied demonstrated the seasonal variations in raindrop size distribution (DSD) over different regions in India (Reddy and Kozu, 2003; Harikumar et al., 2009; Jayalakshmi and Reddy, 2014; Harikumar 2016; Das et al., 2017; Lavanya et al., 2019). However, the climatological studies of DSD over orographic regions are limited, especially in the Western Ghats (WGs) region. Despite its orography, the rainfall intensity is less (below 10 mm h$^{-1}$) over WGs (Sasikumar et al., 2007; Das et al., 2017). A few attempts have been made to understand the DSD characteristics in WGs. For example, Konwar et al. (2014) studied the DSD characteristics by fitting three-parameter gamma function during monsoon (JJAS). They observed a bimodal and monomodal DSD during low and high rainfall rates, respectively. However, their study is limited to brightband and non-brightband conditions only. Harikumar (2016) studied the DSD differences between coastal (Kochi) and high altitude (Munnar) stations located in the WGs. He found that the larger drops are more at Munnar that at Kochi for a given rain rate. Das et al. (2017) studied the DSD characteristics during different precipitating systems in the WGs region using disdrometer and Micro Rain Radar measurements. They noticed different Z-R relations for different precipitating systems. Sumesh et al. (2019) studied the DSD differences between mid- (Braemore, 400 m above mean sea level) and high-altitude (Rajamallay, 1820 m above mean sea level) regions in southern WGs during brightband events. They observed bimodal DSD in the mid-altitude station and monomodal DSD in the high-altitude station. However, their study confined to stratiform rain only.**

**The DSD studies are inadequate in the WGs region by considering long-term dataset. This work is the first to analyze the DSD characteristics and plausible dynamic and microphysical processes by considering the monsoon intra-seasonal oscillations (wet and dry spells). The present study brings out the results of a unique opportunity by analyzing a more extensive dataset and considering different phases of monsoon intra-seasonal oscillations in the WGs. With this background, the current study attempt to address the following issues over WGs:**
**1. How do the DSD characteristics vary during wet and dry spells?**
**2. Does the wet and dry spell rainfall have different microphysical origin over the complex terrain of WGs?**
**3. Does the DSD show any diurnal differences like rainfall distribution during wet and dry spells?**

**4. What are the dynamical processes influencing DSD characteristics during wet and dry spells?**
**5. Establish the best fit for µ-Λ relationships during wet and dry spells.**

**The necessary sentences are added in the revised manuscript.**

**Comment#2**   *Line 118-121: Please add a figure to show the topographic map and location of the disdrometers used in this study.*

**Response**   **The topographic map along with the disdrometer location is added in the revised manuscript.**

**Comment#3**   *Line 190-196: Please explain the selection of temperature and specific humidity at 700 hPa as the data sources of water vapor analysis. Why the wind feature is not considered in the analysis. Also, the spatial resolution of ERA-Interim is 0.25 x 0.25 deg, and the disdrometer data is site-based. How to fill and explain the uncertainty of the spatial resolution between ERA-Interim and in-situ disdrometer.*

**Response**   **To understand the dynamical mechanisms that influence DSD in wet and dry spells, specific humidity, temperature, horizontal, and vertical winds are analyzed at 850 hPa during monsoon of 2012-2015 using ERA-Interim data. As the observational site is at a height of ~ 1.4 km above mean sea level, the temperature, and moisture availability at 850 hPa will provide the information about growth of active convection over the study region.**
**We agree with the reviewer that the disdrometer provides DSD at the surface over a specific location, whereas ERA-Interim data provides the background atmospheric conditions at 0.25 × 0.25 degree grid resolution. It should be noted here that the authors aim to understand the large-scale (dynamical) features responsible for the observed DSD differences only. The authors did not intend to quantify the effect of atmospheric conditions on the DSD differences in wet and dry spells. In addition, if there is any uncertainty arises due to different spatial resolutions between ERA-Interim and in-situ disdrometer then that will be present for both wet and dry spells, and hence will not change the conclusions of this work.**

**Comment#4**   *Line 374-378: The precipitation classification algorithm needs to be further clarified as the Bringi et al. (2003) approach has been improved since 2003.*

**Response**   **Several rain classification schemes proposed in the literature using different instruments, like, disdrometer, radar, profiler (Bringi et al., 2003; Thompson et al., 2015; Krishna et al., 2016; Das et al., 2017; Dolan et al., 2018; Harikumar et al., 2019). In this study, the rainfall at the ground is classified as stratiform and convective based on the criterion proposed by Bringi et al. (2003). Even though several other classification schemes available in the literature, it is the most widely used classification criterion for stratiform and convective rainfall. For instance, several past and recent studies (Marzano et al., 2010; Chen et al., 2013; Tnag et al., 2014; Wen et al., 2016; Suh et al., 2016; Wu and Liu, 2017; Seela et al., 2017; 2018) used the same Bringi et al. (2003)**

criterion for the classification of precipitation systems. There are slight differences among different classification schemes, which leads to small differences in DSD characteristics, and hence the choice of different classification schemes is subjective. To the best of author's knowledge, Bringi et al. (2003) criterion didn't have any limitation/drawbacks for analyzing the DSD spectra in the WGs regions. So the authors strongly believe that Bringi et al. (2003) criterion can effectively be used to classify stratiform and convective rain types. As the present study intends to understand the DSD differences between convective and stratiform (rain which does not come under the convective category) rain systems, we adopted the well-known Bringi et al. (2003) criterion.

**Comment#5** *Line 564-567: The explanation of feedback mechanism during the wet and dry spells are weak. Although Fig.10 shows the temperature and specific humidity patterns during wet and dry spells, lacks of enough quantitative analysis. Please give more evidences to support the feedback mechanism.*

**Response** The differences in $D_m$ during wet and dry spells might occurred either at the cloud formation stage and/or during descent of the precipitation particles to ground. The microphysical and dynamical processes during descent of the precipitation particles are responsible for the spatial-temporal variability of $D_m$ (Rosenfeld and Ulbrich, 2003). The dominant dynamical processes that affect the $D_m$ are updrafts/downdrafts, and advection by horizontal winds. To understand the dynamical mechanisms leading to different microphysical processes in wet and dry periods, we have analyzed temperature, specific humidity, horizontal and vertical winds for 2012-2015 monsoon period over WGs. Figure 1 shows the anomalies in specific humidity (kg kg$^{-1}$, shading), temperature (K, solid contour line), and horizontal winds (vectors) at 850 hPa derived from ERA-Interim reanalysis dataset. This level is chosen, as the temperature anomaly and the availability of moisture at this level aid the growth of active convection. The daily 0000 UTC ERA-Interim fields for ten years (2006-2015) are considered to find anomalies. The seasonal averages are calculated for different atmospheric parameters and the anomalies are calculated as the difference between wet/dry period mean and seasonal mean. Here, positive anomalies in specific humidity (temperature) represent increase in moisture content (heating), and negative anomaly represents decrease in specific humidity (cooling). It is observed that the temperature is cooler over west coast of India (including the study region) in wet spells compared to that in the periods. The figure also shows that the anomalous winds are maritime, and continental during wet and dry spells, respectively. The anomalous winds coming from ocean brings more moisture (positive anomalies in specific humidity) over WGs during wet spells. Whereas, the anomalous winds coming from continent brings dry (negative anomalies in specific humidity) air during dry spells. The thermal gradient between WGs and surrounding regions and the availability of more moisture favours active convection in the wet spells. Whereas, positive temperature anomalies in dry spell can lead to the evaporation of raindrops, which subsequently can break the drops, thereby leading to lesser diameter drops in the dry

spell.

[Figure]

**Fig. 1: Spatial distribution of anomalies in specific humidity (kg kg$^{-1}$, shading), temperature (K, solid contour line), and horizontal winds (vectors) at 850 hPa during wet and dry spells of the monsoon 2012-2015. Here, positive anomalies in specific humidity (temperature) represents increase in moisture content (heating), and negative anomaly represents decrease in moisture (cooling). The black dot represents the observational site.**

To understand the effect of updrafts/downdrafts on the observed variability in $D_m$ distribution, the profile of omega (vertical motion in pressure coordinate) around the study region (17-18$^o$N and 73-74$^o$N) is analysed and is shown in Figure 2. Here, negative values of omega represents updrafts and vice-versa. The mean vertical winds are negative in wet spells indicating updrafts. Whereas the mean vertical winds are small and positive indicating downdrafts during dry spells. The updrafts do not allow the smaller drops to fall, which are carried aloft, where they can fall out later. Hence, the smaller drops have enough time to grow by the collision-coalescence process, to form mid- or large-size drops. Therefore, the mid- or large-size drops increases at the expense of smaller drops, which leads to larger $D_m$ values during wet spells. Whereas the downward flux of raindrops increases due to downdrafts, which causes smaller drops reaching the surface. The large density of smaller drops decreases $D_m$ value during dry spells.

[Figure]

**Fig. 2: The mean profile of vertical velocity during wet and dry spells.**

**This shows that the dynamical mechanisms underlying the microphysical processes are different, which causes the difference in observed DSD characteristics during wet and dry spells.**
**The above analysis is added in the revised manuscript.**

**References:**

Bringi, V. N., Chandrasekar, V., Hubbert, J., Gorgucci, E., Randeu, W. L., and Schoenhuber M.: Raindrop size distribution in different climatic regimes from disdrometer and dual-polarized radar analysis, J. Atmos. Sci., 60, 354–365, 2003.

Chen, B., Yang, J. and Pu, J.: Statistical Characteristics of Raindrop Size Distribution in the Meiyu Season Observed in Eastern China, J. Meteorol. Soc. Japan, 91(2), 215–227 [online] Available from: https://ci.nii.ac.jp/naid/40019636893/en/, 2013.

Das, S. K., Konwar, M., Chakravarty, K. and Deshpande, S M.: Raindrop size distribution of different cloud types over the Western Ghats using simultaneous measurements from Micro-Rain Radar and disdrometer, Atmos. Res., 186, 72–82, doi:http://dx.doi.org/10.1016/j.atmosres.2016.11.003, 2017.

Dolan, B., Fuchs, B., Rutledge, S. A., Barnes, E. A. and Thompson, E. J.: Primary Modes of Global Drop Size Distributions, J. Atmos. Sci., 75(5), 1453–1476, doi:10.1175/JAS-D-17-0242.1, 2018.

Harikumar, R.: Discernment of near-oceanic precipitating clouds into convective or stratiform based on Z–R model over an Asian monsoon tropical site, Meteorology and Atmospheric Physics, 132:377–390, 2020.

Harikumar, R., Sampath, S. and Kumar, V. S.: An empirical model for the variation of rain drop size distribution with rain rate at a few locations in southern India, Adv. Sp. Res., 43(5), 837–844, doi:10.1016/j.asr.2008.11.001, 2009.

Harikumar, R.: Orographic effect on tropical rain physics in the Asian monsoon region, Atmos. Sci. Lett., 17, 556-563, doi:10.1002/asl.692, 2016.

Jayalakshmi, J., and Reddy, K. K.: Raindrop size distributions of southwest and northeast monsoon heavy precipitations observed over Kadapa (14°40N, 78°820E), a semi-arid region of India. Current Science, 107(8), 1312–1320, 2014.

Konwar, M., Das, S. K., Deshpande, S. M., Chakravarty, K. and Goswami, B. N.: Microphysics of clouds and rain over the Western Ghat, J. Geophys. Res. Atmos., 119(10), 6140–6159, doi:10.1002/2014JD021606, 2014.

Krishna, U. V. M., Reddy, K. K., Seela, B. K., Shirooka, R., Lin, P.-L. and Pan, C.-J.: Raindrop size distribution of easterly and westerly monsoon precipitation observed over Palau islands in the Western Pacific Ocean, Atmos. Res., 174–175, 41–51, doi:https://doi.org/10.1016/j.atmosres.2016.01.013, 2016.

Lavanya, S., Kirankumar, N. V. P., Aneesh, S., Subrahmanyam, K. V and Sijikumar, S.: Seasonal variation of raindrop size distribution over a coastal station Thumba : A quantitative analysis, Atmos. Res., 229(June), 86–99, doi:10.1016/j.atmosres.2019.06.004, 2019.

Marzano, F. S., Cimini, D. and Montopoli, M.: Investigating precipitation microphysics using ground-based microwave remote sensors and disdrometer data, Atmos. Res., 97(4), 583–600, doi:https://doi.org/10.1016/j.atmosres.2010.03.019, 2010.

Reddy, K. K and Kozu, T.: Measurements of raindrop size distribution over Gadanki during south-west and north-east monsoon, Indian J. Radio & Space Phys, 32(October), 286–295, 2003.

Rosenfeld, D. and Ulbrich, C. W.: Cloud Microphysical Properties, Processes, and Rainfall Estimation Opportunities BT - Radar and Atmospheric Science: A Collection of Essays in Honor of David Atlas, edited by R. M. Wakimoto and R. Srivastava, pp. 237–258, American Meteorological Society, Boston, MA., 2003.

Sasi Kumar, V., Sampath, S., Vinayak, P. V. S. S. K. and Harikumar, R.: Rainfall intensity characteristics at coastal and high altitude stations in Kerala, J. Earth Syst. Sci., (5), 451–463, 2007.

Seela, B. K., Janapati, J., Lin, P.-L., Reddy, K. K., Shirooka, R. and Wang, P. K.: A Comparison Study of Summer Season Raindrop Size Distribution Between Palau and Taiwan, Two Islands in Western Pacific, J. Geophys. Res. Atmos., 122(21), 11,711-787,805, doi:10.1002/2017JD026816, 2017.

Seela, B. K., Janapati, J., Lin, P.-L., Wang, P. K. and Lee, M.-T.: Raindrop Size Distribution Characteristics of Summer and Winter Season Rainfall Over North Taiwan, J. Geophys. Res. Atmos., 123(20), 11,602-611,624, doi:10.1029/2018JD028307, 2018.

Suh, S.-H., You, C.-H. Lee, D.-I.: Climatological characteristics of raindrop size distributions in Busan, Republic of Korea, Hydrol. Earth Syst. Sci., 20, 193–207, 2016.

Sumesh, R. K., Resmi, E. A., Unnikrishnan, C. K., Jash, D., Sreekanth, T. S., Resmi, M. C. M., Rajeevan, K., Nita, S. and Ramachandran, K. K.: Microphysical aspects of tropical rainfall during Bright Band events at mid and high-altitude regions over Southern Western Ghats , India, Atmos. Res., 227(March), 178–197, doi:10.1016/j.atmosres.2019.05.002, 2019.

Tang, Q., Xiao, H., Guo, C. and Feng, L.: Characteristics of the raindrop size distributions and their retrieved polarimetric radar parameters in northern and southern China, Atmos. Res., 35–136, 59-75, 2014.

Thompson, E. J., Rutledge, S. A., Dolan, B. and Thurai, M.: Drop Size Distributions and Radar Observations of Convective and Stratiform Rain over the Equatorial Indian and West Pacific Oceans, J. Atmos. Sci., 72(11), 4091–4125, doi:10.1175/JAS-D-14-0206.1, 2015.

Wen, L., Zhao, K., Zhang, G., Xue, M., Zhou, B., Liu, S. and Chen, X.: Statistical characteristics of raindrop size distributions observed in East China during the Asian summer monsoon season using 2-D video disdrometer and Micro Rain Radar data, J. Geophys. Res. Atmos., 121(5), 2265–2282, doi:10.1002/2015JD024160, 2016.

Wu, Y. and Liu, L.: Statistical Characteristics of Raindrop Size Distribution in the Tibetan Plateau and Southern China, Adv. Atmos. Sci., 34, 727–736, 2017.

---

## Author Response (AR2)

**Editor comment on "Statistical characteristics of raindrop size distribution over Western Ghats of India: wet versus dry spells of Indian Summer Monsoon" by Uriya Veerendra Murali Krishna et al.**

| | |
|---|---|
| **General Comment** | *I find a substantial improvement in the revised MS and the authors have the referee concerns. Thank you for revising the MS. The MS is now ready for publication in ACP, but please consider the below-given suggestions too.* |
| **Response** | **We are indebted to the editor for his valuable and thoughtful comments on the manuscript. We went through all the editor comments and suggestions and implemented the same in the revised manuscript. Point-to-point clarifications for the comments and how we have addressed each recommendation is provided below.** |

**Specific Comments**

| | |
|---|---|
| **Comment. 1** | *L1: analysed for* |
| **Response** | **Thanks. Corrected.** |
| | |
| **Comment. 2** | *L3: ISM periods in 2012-2015* |
| **Response** | **Corrected.** |
| | |
| **Comment. 3** | *L5: not very prominent* |
| **Response** | **Corrected.** |
| | |
| **Comment. 4** | *L6: ". However, "* |
| **Response** | **Corrected.** |
| | |
| **Comment. 5** | *L6: What are the dynamical processes? Write at least one , like "dynamical processes such as …"* |
| **Response** | **The appropriate sentence is corrected as "However, the underlying dynamical parameters such as moisture availability, vertical wind, etc. cause the differences in DSD characteristics."** |
| | |
| **Comment. 6** | *L11: write the main differences here.* |
| **Response** | **The necessary sentence is added.** |
| | |
| **Comment. 7** | *L44: Henceforth,* |
| **Response** | **Corrected** |
| | |
| **Comment. 8** | *L50: write "," instead of along with* |
| **Response** | **Corrected.** |
| | |
| **Comment. 9** | *L54: not elements but "cells"* |
| **Response** | **The sentence is modified as "All these features promote the anomalous south-westerlies, which enhances convective activity over WGs."** |
| | |
| **Comment. 10** | *L54: "which, enhances convective activity"* |
| **Response** | **The sentence is modified as "All these features promote the anomalous south-westerlies, which enhances convective activity over** |

WGs."

**Comment. 11** *L56: What atmospheric condition?*

**Response** The sentences is modified as "… positive geopotential height anomalies, positive OLR anomalies, and negative precipitable water anomalies are observed during the dry spells, which suppress the convective activity in the Arabian Sea, and hence little to no rain is seen over WGs during dry periods"

**Comment.1 2** *L60-73: One important reference missing in this section is Varikoden et al. doi.org/10.1007/s00382-018-4397-7. They discuss how the rainfall changed over WG in the past decades, and the differences in rainfall trends in the northern and southern WG.*

**Response** The importance of spatial variability of rainfall in WGs is added in the introduction with appropriate references.

**Comment. 13** *L67: "region, and found that the larger drops are relatively higher at Munnar"*

**Response** Thanks. Corrected.

**Comment. 14** *L71-72: can you write the mid and high altitudes in km here?*

**Response** Sumesh et al. (2019) studied the DSD differences between mid-(Braemore, 0.4 km above mean sea level) and high-altitude (Rajamallay, 1.8 km above mean sea level) regions in southern WGs during brightband events.

**Comment. 15** *L80: Can you write the answers to these five points in Abstract?*

**Response** Thanks for the suggestion. The key findings of this study are added in the abstract.

**Comment. 16** *L112: "small drops do not significantly affect"*

**Response** Corrected.

**Comment. 17** *L211: droplet growth*

**Response** Corrected.

**Comment. 18** *L212: "indicates drop breakup and evaporation ..."*

**Response** Corrected.

**Comment. 19** *L275: "are available"*

**Response** Corrected.

**Comment. 20** *Conclusions: There are eight points and some are discussed in the text and are not new. Please give the new information from your study here.*

**Response** Thanks. The conclusion section is summarized with major findings from the present observations.

**Comment. 21** *Figure 8, 9, and 10: Make the y-axis uniform with respect to the values (give a common range) so that we can compare them. Also, combine all three to single figure with sub-plots.*

**Response**  **Thanks. Figure 8, 9, 10 are combined and shown with Figure 8 in the manuscript. The y-axis range made uniform in figure 8.**

**Comment. 22**  *Figure 16: is should be introduced in the last paragraph of the discussion, not in conclusion.*

**Response**  **Thanks. Figure 16 is added in the discussion section as Figure 13 in the revised manuscript.**